# NeuroLM: A Universal Multi-task Foundation Model for Bridging the Gap between Language and EEG Signals

**Wei-Bang Jiang**[1][*]**, Yansen Wang**[2]**, Bao-Liang Lu**[1]**, Dongsheng Li**[2]

[1]Shanghai Jiao Tong University    [2]Microsoft Research Asia

`{935963004,bllu}@sjtu.edu.cn,{yansenwang,dongsli}@microsoft.com`
`https://github.com/935963004/NeuroLM`

## Abstract

Recent advancements for large-scale pre-training with neural signals such as electroencephalogram (EEG) have shown promising results, significantly boosting the development of brain-computer interfaces (BCIs) and healthcare. However, these pre-trained models often require full fine-tuning on each downstream task to achieve substantial improvements, limiting their versatility and usability, and leading to considerable resource wastage. To tackle these challenges, we propose NeuroLM, the first multi-task foundation model that leverages the capabilities of Large Language Models (LLMs) by regarding EEG signals as a foreign language, endowing the model with multi-task learning and inference capabilities. Our approach begins with learning a text-aligned neural tokenizer through vector-quantized temporal-frequency prediction, which encodes EEG signals into discrete neural tokens. These EEG tokens, generated by the frozen vector-quantized (VQ) encoder, are then fed into an LLM that learns causal EEG information via multi-channel autoregression. Consequently, NeuroLM can understand both EEG and language modalities. Finally, multi-task instruction tuning adapts NeuroLM to various downstream tasks. We are the first to demonstrate that, by specific incorporation with LLMs, NeuroLM unifies diverse EEG tasks within a single model through instruction tuning. The largest variant NeuroLM-XL has record-breaking 1.7B parameters for EEG signal processing, and is pre-trained on a large-scale corpus comprising approximately 25,000-hour EEG data. When evaluated on six diverse downstream datasets, NeuroLM showcases the huge potential of this multi-task learning paradigm.

## 1 Introduction

Electroencephalogram (EEG) signals have become a cornerstone in the development of brain-computer interfaces and healthcare domains, offering a non-invasive solution to capture the electrical activity of the brain. EEG measures the voltage fluctuations resulting from ionic current flows within the neurons of the brain, providing real-time insights into brain function and neural dynamics. This capability makes EEG an invaluable tool for creating interfaces that enable direct communication between the brain and external devices. EEG is advantageous due to its high temporal resolution, cost-effectiveness, and portability, and has been significantly enhanced by advanced computational methods. Therefore, a wide range of applications have been utilizing EEG signals, including but not limited to human emotion recognition (Jenke et al., 2014),

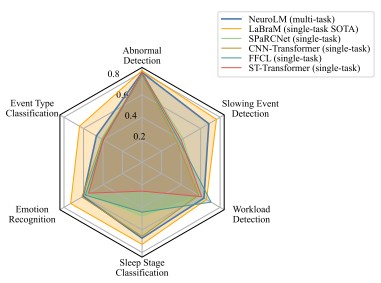

Figure 1: Comparison on six tasks.

body motor imaginary (Tabar & Halici, 2016), automatic sleep stage classification (Supratak et al., 2017), seizure epilepsy detection (Alotaiby et al., 2014), and fatigue detection (Gao et al., 2019).

---

[*]Work done during Wei-Bang's internship at Microsoft Research Asia. Correspondence to Yansen Wang.

While EEG signals are popular among researchers, they have several disadvantages, including the low signal-to-noise ratio, inherent nonstationarity, as well as diverse configurations in EEG data collection. Besides, there is a lack of sufficient and consistent EEG data. These challenges complicate the extraction of universal EEG representations. To overcome these problems, several studies have proposed methods compatible with diverse EEG configurations to learn effective and generic representations. For example, Yang et al. (2023a) introduce a Biosignal Transformer (BIOT), which unifies various EEG data by tokenizing channels into fix-length segments with channel and relative position embeddings for preserving spatio-temporal features. Jiang et al. (2024) advance this approach by proposing a neural tokenizer to pre-train LaBraM by masked neural code prediction with 2,500 hours of EEG data, thus achieving state-of-the-art (SOTA) performance on various downstream tasks. Although these methods effectively address the aforementioned challenges, they still require individual fine-tuning on each downstream dataset to obtain impressive improvement. Despite increasing model size and employing large-scale unsupervised pre-training to learn generic representations, such adaptation confines the fine-tuned model to perform only a single task. Moreover, this task-specific fine-tuning demands substantial computational and storage resources.

Over the past few years, the advent of Large Language Models has brought remarkable progress and demonstrated extraordinary emergent abilities (Brown et al., 2020; Touvron et al., 2023). The development of LLMs has given rise to Multimodal Large Language Models (MLLMs) (Achiam et al., 2023; Liu et al., 2023), which unleash the potential of powerful LLMs to perform multimodal tasks. MLLMs typically integrate a modality-specific encoder, pre-aligned with text embeddings, into off-the-shelf LLMs. Inspired by MLLMs, we unveil a new direction of integrating multiple EEG tasks into a unified model by incorporating EEG signals into existing LLMs. However, there are some challenges in harnessing LLMs to understand EEG patterns, comprising:

**1) EEG-text embedding alignment.** Aligning EEG and text embeddings presents a great challenge. Unlike vision-language models which benefit from numerous high-quality image-text pairs, there are no established EEG-text pairs available due to the difficulty of extracting semantic information from a given EEG segment.

**2) Effective Representation learning with LLMs.** Mainstream methods employ masked EEG modeling to effectively extract representations for EEG signals. When integrating LLMs, how to learn generic information within the LLM paradigm remains an unsolved issue.

**3) Unified multi-task learning with various EEG tasks.** Integrating multiple EEG tasks into a unified model is complex due to the diversity and specificity of different tasks. Developing a model that can seamlessly handle various tasks without compromising performance on any individual task is a major challenge.

In light of the aforementioned challenges, we propose NeuroLM, a universal multi-task foundation model for EEG signal processing. NeuroLM builds upon the compatibility with diverse EEG formats established by LaBraM, and it is pre-trained on a large-scale dataset comprising approximately 25,000 hours of EEG data. The training of NeuroLM involves three stages. First, a text-aligned neural tokenizer is trained using vector-quantized temporal-frequency prediction to encode continuous EEG signals into discrete codes from a neural codebook, with adversarial training employed to align the EEG and text spaces. Next, the VQ encoder of the neural tokenizer is frozen to extract compact embeddings, which serve as input for a LLM. To enable the LLM to learn causal EEG representations, we propose multi-channel autoregressive pre-training, which mimics autoregressive language modeling but is tailored for multi-channel EEG signals. Finally, we elaborate instructions for various downstream datasets and employ multi-task instruction tuning to empower NeuroLM for multi-task learning. Experiments on six different tasks, encompassing abnormal detection, event type classification, emotion recognition, sleep stage classification, cognitive workload prediction, and slowing type classification, demonstrate NeuroLM's superiority in multi-task learning and inference. To the best of our knowledge, we are the first to introduce instruction tuning to enable multi-task learning and inference in the field of EEG signal processing. The highlights are summarized as follows:

**1) Text-aligned neural tokenizer embeddings.** We introduce a text-aligned neural tokenizer that effectively bridges the gap between EEG and text data. This tokenizer uses vector-quantized temporal-frequency prediction to convert EEG signals into discrete codes, facilitating the alignment of EEG and text embeddings through adversarial training. This alignment is crucial for leveraging the strengths of LLMs in understanding and processing EEG data.

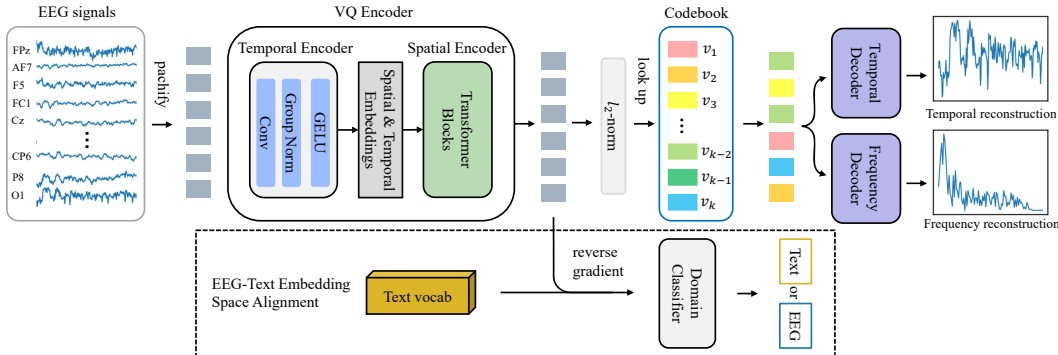

Figure 2: The architecture design of text-aligned neural tokenizer training. The neural tokenizer is trained by reconstructing both temporal and frequency domain of input EEG signals to discretize them into discrete neural tokens. To align EEG and text embedding space, we utilize a domain classifier through adversarial training.

**2) Large-scale multi-channel autoregressive pre-training.** NeuroLM employs multi-channel autoregression, enabling the model to learn causal representations across different EEG channels. Pre-training on 25,000 hours of EEG data ensures that NeuroLM captures a wide range of neural patterns, enhancing its ability to generalize across diverse EEG tasks.

**3) Joint multi-task tuning and inference.** We pioneer the use of joint multi-task tuning and inference for EEG. By elaborating specific instructions for various downstream tasks and employing multi-task instruction tuning, NeuroLM is capable of performing multiple tasks within a single model. This not only improves efficiency by reducing the need for individual fine-tuning for each task but also ensures high performance across a spectrum of applications.

## 2 METHOD

In this section, we elaborate our design of NeuroLM. We first train a neural tokenizer by vector-quantized temporal-frequency prediction. Whereafter, the VQ encoder of the tokenizer will serve to encode EEG signals into embeddings aligned with text space, and the EEG embeddings will be seamlessly used as input to Large Language Models.

Given multi-channel EEG signals $X \in \mathbb{R}^{C \times T}$, where $C$ denotes the number of channels and $T$ denotes total timestamps. An EEG sample is formulated as $x \in \mathbb{R}^{C \times L}$, where $L$ is the window size, resulting in a total number of $\lfloor \frac{T}{L} \rfloor$ samples. We pachify the EEG samples into non-overlap patches $x = \{x_{ij} \in \mathbb{R}^P | i = 1, ..., C, j = 1, ..., N\}$. Let $P$ is patch size and $N = \lfloor \frac{L}{P} \rfloor$.

### 2.1 TEXT-ALIGNED NEURAL TOKENIZER TRAINING

To incorporate EEG into off-the-shelf Large Language Models, we first need to encode EEG signals into embeddings whose space is well-aligned with text embedding space. VQ-VAE (Van Den Oord et al., 2017) is a good choice that maps continuous signals to discrete tokens while preserving the key information. Our text-aligned neural tokenizer basically follows the well-established neural tokenizer of LaBraM (Jiang et al., 2024) with some improvements. Vector-quantized temporal-frequency prediction is utilized to train the text-aligned neural tokenizer, as illustrated in Figure 2.

**Neural Tokenizer.** The neural tokenizer is composed of several vital components: VQ encoder, codebook, temporal/frequency decoder, and domain classifier. The codebook $\mathcal{V} \in \mathbb{R}^{K \times D}$ contains $K$ discrete $D$-dimension embeddings. Let $h_i$ denote the patch representations derived from the VQ encoder. We find the nearest codes of each $h_i$ from codebook embeddings $\{v_i | i = 1, ..., K\}$:

$$z_i = \arg \min_j \|\ell_2(h_i) - \ell_2(v_i)\|_2, \tag{1}$$

where $j \in \{1, ..., K\}$ and $\ell_2$ normalization is employed so that the above distance is equivalent to cosine similarity. Consequently, an EEG sample is tokenized to $z = [z_1, ..., z_N]$.

**Temporal-frequency Prediction.** We propose to predict both original signals and the frequency magnitude to capture the temporal and frequency domains of EEG signals. This differs from LaBraM which regresses the Fourier amplitude and phase since we observe that reconstructing the phase contributes minor to neural tokenizer training. We apply the Discrete Fourier Transform (DFT) on an EEG patch $x_{i,j} = [x[1], x[2], ..., x[P]]$ of channel $i$ and time $j$, and transform the equation using Euler's formula as follows

$$\tilde{x}_{i,j}^m = \sum_{n=1}^{M} x[n] \cos(\frac{2\pi}{M}mn) - \boldsymbol{j}x[n]\sin(\frac{2\pi}{M}mn). \tag{2}$$

where $m \in [1, N]$ and $\boldsymbol{j}$ is the imaginary unit. Accordingly, we calculate the frequency magnitude as $f^m = \sqrt{Re(\tilde{x}_{i,j}^m)^2 + Im(\tilde{x}_{i,j}^m)^2}$, where $Re$ and $Im$ represent the real and imaginary parts of a complex number. For stable convergence, we adopt z-score normalization to the magnitude within a sample.

After being quantized to the codebook embeddings, we feed the normalized neural embeddings $[\ell_2(z_1), ..., \ell_2(z_N)]$ into two separate decoders. Let $o_i^t$ and $o_i^f$ stand for the output of a temporal decoder and a frequency decoder, respectively. The optimizing target for the codebook learning is

$$\mathcal{L}_1 = \sum_{\boldsymbol{x}\in\mathcal{D}} \sum_i \underbrace{\|o_i^t - x_i\|_2^2 + \|o_i^f - f_i\|_2^2}_{\text{reconstruction loss}} + \underbrace{\|\boldsymbol{sg}(\ell_2(h_i)) - \ell_2(v_{z_i})\|_2^2}_{\text{codebook loss}} + \underbrace{\|\ell_2(h_i) - \boldsymbol{sg}(\ell_2(v_{z_i}))\|_2^2}_{\text{commitment loss}}, \tag{3}$$

where $\mathcal{D}$ represents the whole dataset and $\boldsymbol{sg}$ denotes the stop-gradient operator that is identical during forward computation and has zero partial derivatives.

**EEG-text Embedding Space Alignment.** Current vision-language models usually utilize pre-trained CLIP-like (Radford et al., 2021) image encoders which are trained by large-scale image-text pairs and thus are embedding-wise well-aligned with text. However, when considering EEG, there are much more challenges to align EEG with text: 1) EEG signals contain complicated cognitive and non-cognitive information, which is hard to be described by human language accurately and thoroughly. For example, an EEG segment can not only contain one person's emotion and mental states, but also represent the body movement and medical normality. 2) The labeled EEG data available to construct EEG-text pair are very limited. Therefore, we propose to align EEG with text space-wise instead of embedding-wise.

We introduce a domain classifier $\mathcal{C}$ to predict whether the embeddings are from EEG or text. During the codebook learning, we also feed some text embeddings from LLMs to train the domain classifier. A gradient reverse layer (Ganin et al., 2016) is added after the VQ encoder to confuse the domain classifier. Hence, the embeddings from the VQ encoder fall into the same space of text embeddings. Consequently, the training objective for text-aligned neural tokenizer training is defined as

$$\min \mathcal{L}_1 + \lambda \sum_i d_i \log \mathcal{C}(h_i), \tag{4}$$

where $d_i$ is the label of EEG or text domain and $\lambda = \frac{2}{1+e^{-10t/T}} - 1$ is a scaling factor that gradually changes from 0 to 1.

**VQ Encoder Architecture.** We briefly introduce the architecture of the VQ encoder as it is almost the same as LaBraM. The temporal encoder and spatial encoder are two pivotal parts of the VQ encoder. The temporal encoder contains several blocks of 1-D convolution which aims to extract temporal features in each EEG patch. After that, learnable temporal and spatial embeddings are added according to the standard 10-20 international system to inject both time and channel information. Finally, the spatial encoder composed of vanilla Transformer blocks (Vaswani et al., 2017) learns interaction among patches.

## 2.2 MULTI-CHANNEL AUTOREGRESSIVE PRE-TRAINING

Before passing EEG data into Large Language Models, we freeze the VQ encoder and first use it to encode input EEG data to EEG tokens that are aligned with the text space. After that, we load a pre-trained Large Language Model and enlarge the text vocabulary with the learned EEG codebook.

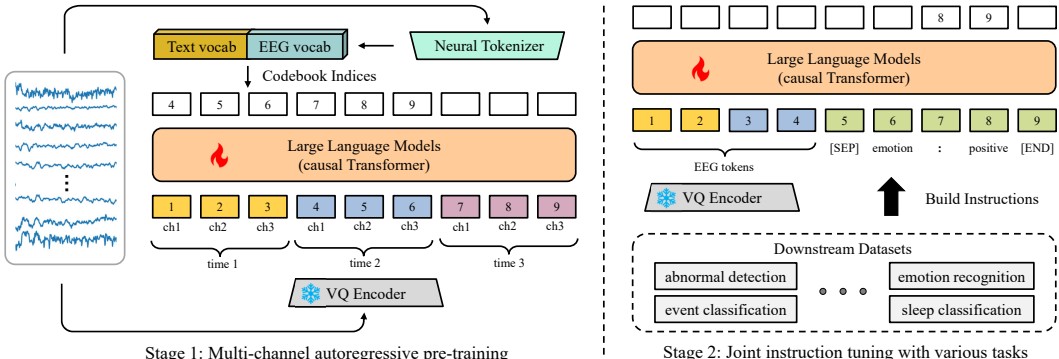

Figure 3: Schematic of NeuroLM training. **Left**: We first pre-train NeuroLM via multi-channel autoregression with EEG tokens output by the frozen VQ encoder. **Right**: The multi-task instruction tuning enables NeuroLM to perform various BCI tasks within a single model.

The EEG tokens are added with reused temporal embeddings from the LLM and new spatial embeddings. As shown in Figure 3, NeuroLM is then trained through multi-channel autoregression, that is, predicting the next EEG tokens based on visible EEG tokens, to endow the model with the capability of learning special patterns of EEG causal relationship. In our experiments, the multi-channel autoregressive pre-training contributes to the performance of multi-task instruction tuning.

**Formulation.** Consider a sequence of EEG tokens $h = \{h_{ij}|i = 1, ..., C, j = 1, ..., T\}$ where $i$ denotes the channel and $j$ denote the time, and their corresponding indices of the merged text and EEG vocabulary $I = \{I_{ij}|i = 1, ..., C, j = 1, ..., T\}$ derived from the neural tokenizer. Unlike language that can be predicted token by token intuitively, EEG signals are of various configurations, thus it is impracticable to directly predict EEG tokens one by one. We propose a multi-channel autoregressive strategy to adopt the idea of autoregression on EEG. The basic idea is that each token of a specific channel predicts the next token of the same channel, which can be formulized as

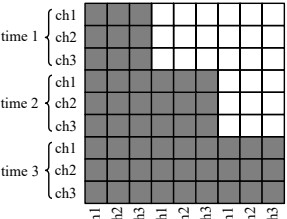

Figure 4: The stair-stepping mask. Each row indicates attention masks for an EEG token.

$$p(I_{11}, I_{12}, ..., I_{CT}) = \prod_{t=1}^{T} p(I_{1n}, I_{2n}, ..., I_{Cn}|h_{11}, h_{12}, ..., h_{C(t-1)}). \quad (5)$$

Therefore, the objective for multi-channel autoregressive pre-training is to optimize model parameters by maximizing $p(h_{1t}, h_{2t}, ..., h_{Ct}|h_{11}, h_{12}, ..., h_{C(t-1)})$ throughout all EEG data.

For implementation, we define stair-stepping masks where each EEG token is able to observe tokens of all channels from its current and previous time step. Figure 4 illustrates the design of our stair-stepping mask. Dark cells indicate that the elements should take part in attention.

**Theory Analysis.** We interpret the multi-channel autoregressive pre-training from the view of a variational autoencoder (Kingma & Welling, 2014). Let $x$ denote the original EEG signals, $y$ denote the temporal-frequency target of $x$, and $\hat{x}$ be the EEG tokens to be predicted. Assume that EEG signals $x$ can be generated by a random process with a latent variable $\mathbf{z}$. We use $q_\phi(\mathbf{z}|x_i)$ to denote the VQ encoder encoding EEG signals into discrete neural codes, $p_\psi(y_i|z_i)$ to stand for the temporal and frequency decoder reconstructing temporal-frequency domain from encoded neural codes, and $p_\theta(\mathbf{z}|x_i)$ to represent multi-channel autoregressive pre-training. Consider the log-likelihood $p(y|x)$ and its evidence lower bound (ELBO), involving predicting the temporal-frequency domain of the EEG signals from the next time point:

$$\sum_{i=1}^{N} \log p(y_i|x_i) \geq -\sum_{i=1}^{N} (\mathbb{E}_{z_i \sim q_\phi(\mathbf{z}|x_i)}[-\log p_\psi(y_i|z_i)] + KL(q_\phi(\mathbf{z}|x_i), p_\theta(\mathbf{z}|x_i))), \quad (6)$$

where the first term is the reconstruction loss and the second term is Kullback-Leibler divergence between $q$ and EEG-text conditional prior. Our training paradigm encompasses two-stage learning

processes: 1) The neural tokenizer is optimized by minimizing the reconstruction loss. 2) A LLM learns the prior $p_\theta$ by minimizing $KL$ loss with $q_\phi$ and $p_\psi$ fixed. The sequence $z_i$ can be sampled from $q_\phi(\mathbf{z}|x_i)$ or one-point distribution $z_i = \arg\max_z q_\phi(\mathbf{z}|x_i)$ where we choose the latter for simplicity. In this case, $z_i$ is from the codebook $\mathcal{V}$ and $z_i = [z_{i,1}, ..., z_{i,T}]$. Therefore, Equation 6 can be rewritten as

$$-\sum_{i=1}^{N}(\mathbb{E}_{z_i \sim q_\phi(\mathbf{z}|x_i)}[-\log p_\psi(y_i|z_i)] - \sum_{j=2}^{T} \log p_\theta(z_{i,j}|x_{i,<j})), \quad (7)$$

where the latter term is the negative log-likelihood loss for multi-channel autoregressive pre-training and $z_{i,j}$ denotes latent variables of all channels at time step $j$.

### 2.3 MULTI-TASK INSTRUCTION TUNING

In this stage, we aim to leverage the power of LLMs to integrate different downstream datasets as a whole. Instruction tuning is introduced to handle various downstream tasks, as shown in Figure 3. It is worthwhile to mention that in both multi-channel autoregressive pre-training and multi-task instruction tuning stages, we feed the model a few text data at each iteration to preserve the language modeling capability of LLMs. We build instructions for each downstream dataset and the instruction design can be found in Appendix B. A special token [SEP] is used to concatenate EEG and text instructions, indicating the modality switch. Notably, the loss is only calculated on the answer part of the text to make the prediction more stable. Suppose $x^p$ represents the EEG tokens along with the question part of the instruction (prompt), and $t^a$ represents the answer part of the instruction. Let the sequence length of $t^a$ be $L$, and this procedure can be written as

$$p(t^a|x^p) = \prod_{i=1}^{L} p(t_i^a|x^p, t_{,<i}^a), \quad (8)$$

where $t_{,<i}^a$ is the answer tokens before the current prediction token $t_i^a$.

## 3 EXPERIMENTS

### 3.1 DOWNSREAM DATASETS

We consider six different EEG datasets with highly varied data sizes to comprehensively evaluate NeuroLM, where the detailed information is listed in Table 1: 1) **TUAB** (Harati et al., 2015) (abnormal detection): This dataset contains EEG records that are classified as clinically normal or abnormal. 2) **TUEV** (Harati et al., 2015) (event type classification): This corpus contains six events involving periodic lateralized epileptiform discharge, generalized periodic epileptiform discharge, spike and/or sharp wave discharges, artifact, and eye movement. 3) **SEED** (Zheng & Lu, 2015) (emotion recognition): There are 3 emotions (positive, negative, and neutral) elicited by videos from 15 subjects. There are 15 trials in each session and each subject underwent 3 sessions. 4) **HMC** (Alvarez-Estevez & Rijsman, 2021) (sleep stage classification): HMC was developed for automatic sleep scoring, involving 5 sleep stages (wake, NREM-1, NREM-2, NREM-3, REM) from 151 subjects. 5) **Workload** (Zyma et al., 2019) (cognitive workload classification): This dataset contains 36 subjects performing serial subtraction. We regard mental workload trials as high workload and the last 60 seconds of the rest EEG as low workload. 6) **TUSL** (von Weltin et al., 2017)

Table 1: Information of datasets used for downstream evaluation.

| Dataset | #Channel | Sampling Rate | Duration | #Sample | Task |
|---|---|---|---|---|---|
| TUAB | 23 | 256 Hz | 10 seconds | 409,455 | Binary classification |
| TUEV | 23 | 256 Hz | 5 seconds | 112,491 | 6-class classification |
| SEED | 62 | 1000 Hz | 4 seconds | 38,475 | 3-class classification |
| HMC | 4 | 256 Hz | 30 seconds | 137,243 | 5-class classification |
| Workload | 19 | 500 Hz | 4 seconds | 2,088 | Binary classification |
| TUSL | 23 | 256 Hz | 10 seconds | 245 | 3-class classification |

(slowing event classification): TUSL aims to differentiate between seizure, slowing, and complex background events.

For the data division, we split each dataset into training, validation, and test sets: 1) **TUAB** and **TUEV**: Since the training and test division is provided by the original datasets, we further divide the training patients into training and validation groups by 80% and 20% randomly. 2) **SEED**: We split total 15 trials into training, validation, and test trials by 9:3:3 according to the chronological order, and merge all sessions into the final training, validation, and test set. 3) **HMC**: The first 100 subjects form the training set while the middle 25 subjects and the last 26 subjects are validation and test sets, respectively. 4) **Workload**: The training, validation, and test sets are derived by subjects from number 0 to 25, number 26 to 30, and number 31 to 35, respectively. 5) **TUSL**: The training, validation, and test sets are splitted by 60%:20%:20%.

## 3.2 EXPERIMENTAL SETUP

**Model Configurations.** NeuroLM is compatible with any causal LLM as its base language model. For simplicity and saving computing resources, we adopt GPT-2 (Radford et al., 2019) as our base language model. Accordingly, NeuroLM has three variants, NeuroLM-B, NeuroLM-L, and NeuroLM-XL, which have 254M, 500M, and 1696M parameters (including the parameters of the VQ encoder), respectively. Unless otherwise noted, NeuroLM refers to NeuroLM-B. Text embeddings which are randomly sampled from GPT-2's vocabulary for each batch, are utilized for EEG-text alignment. The patch size $P$ is set to 200 (1 second), consistent with that of LaBraM. To maintain compatibility with GPT-2, the maximum sequence length (number of patches) is set to 1024. For input samples with sequence lengths shorter than 1024, we pad zeros to ensure the length is equal to 1024 at neural tokenizer training and multi-channel autoregressive pre-training stages. The attention values for these zero paddings will be masked.

**Data Preprocessing.** To eliminate environmental and physiological artifacts from EEG signals, we employ several necessary preprocessing methods. First, we apply a bandpass filter with cutoff frequencies of 0.1 Hz and 75 Hz. To avoid power-line interference, we use a notch filter at 50 Hz or 60 Hz, depending on the geographic region of data collection. Additionally, all signals are resampled to 200 Hz to reduce computational complexity. Given that EEG signal values typically range between -100 $\mu$V to 100 $\mu$V, all values are divided by 100 for normalization.

**Training & Environment Settings.** To facilitate the training of NeuroLM, a huge volumn of data is required. About 25,000 hours of EEG data from multiple public EEG datasets are collected after cleaning and filtering, which are listed in Appendix C. All experiments are conducted on eight NVIDIA A100-80G GPUs with Python 3.11.8 and PyTorch 2.2.2 + CUDA 12.1. For instruction tuning, the results are obtained using the final model after training. Notably, we choose the largest logits as the prediction at evaluation and test instead of beam search which is widely used in current LLMs to obtain stable results. For other scenarios, the baselines are in a single-task manner and trained on individual datasets. Their best models are selected based on the best performance on the validation set, and then evaluated on the test set. The average and standard deviation values are reported using three random seeds to ensure comparable results. Baselines and other detailed hyperparameter settings are provided in Appendix D.

## 3.3 EXPERIMENTAL RESULTS

We present all results in Table 2, 3, and 4. Underlined values represent the best results for single-task methods, while bold values indicate the best results for NeuroLM. Notably, it's important to note that direct comparisons between NeuroLM and the baseline single-task methods are not entirely fair, as the baselines are trained and tested on individual datasets. Although NeuroLM is still a few steps away from the state-of-the-art LaBraM, it achieves performance comparable to most other single-task baselines. The key strength of NeuroLM lies in its unified instruction-tuning, which has the potential to enable generalization to novel tasks or prompts without the need for extensive task-specific fine-tuning. For NeuroLM-L and NeuroLM-XL, the performance is further enhanced on most downstream datasets with larger model capacity. However, the imbalance in data size among different downstream datasets poses a challenge for NeuroLM, as it reaches optimal performance at different training times for different datasets. Additionally, we find that models with more parameters are more prone to overfitting, which might account for the performance degradation observed on

Table 2: Results on TUAB and TUEV.

| Methods | Multi-task | TUAB | | | TUEV | | |
|---|---|---|---|---|---|---|---|
| | | Balanced Acc. | AUC-PR | AUROC | Balanced Acc. | Cohen's Kappa | Weighted F1 |
| SPaRCNet | ✗ | 0.7896±0.0018 | 0.8414±0.0018 | 0.8676±0.0012 | 0.4161±0.0262 | 0.4233±0.0181 | 0.7024±0.0104 |
| ContraWR | ✗ | 0.7746±0.0041 | 0.8421±0.0104 | 0.8456±0.0074 | 0.4384±0.0349 | 0.3912±0.0237 | 0.6893±0.0136 |
| CNN-Transformer | ✗ | 0.7777±0.0022 | 0.8433±0.0039 | 0.8461±0.0013 | 0.4087±0.0161 | 0.3815±0.0134 | 0.6854±0.0293 |
| FFCL | ✗ | 0.7848±0.0038 | 0.8448±0.0065 | 0.8569±0.0051 | 0.3979±0.0104 | 0.3732±0.0188 | 0.6783±0.0120 |
| ST-Transformer | ✗ | 0.7966±0.0023 | 0.8521±0.0026 | 0.8707±0.0019 | 0.3984±0.0228 | 0.3765±0.0306 | 0.6823±0.0190 |
| BIOT | ✗ | 0.7959±0.0057 | 0.8792±0.0023 | 0.8815±0.0043 | 0.5281±0.0225 | 0.5273±0.0249 | 0.7492±0.0082 |
| LaBraM-Base | ✗ | 0.8140±0.0019 | 0.8965±0.0016 | 0.9022±0.0009 | 0.6409±0.0065 | 0.6637±0.0093 | 0.8312±0.0052 |
| NeuroLM-B | ✓ | 0.7826±0.0065 | 0.6975±0.0081 | 0.7816±0.0079 | 0.4560±0.0048 | 0.4285±0.0048 | 0.7153±0.0028 |
| NeuroLM-L | ✓ | 0.7876±0.0034 | 0.7099±0.0034 | 0.7876±0.0034 | 0.4132±0.1235 | 0.4414±0.0996 | **0.7387**±0.0400 |
| NeuroLM-XL | ✓ | **0.7969**±0.0091 | **0.7219**±0.0082 | **0.7884**±0.0194 | **0.4679**±0.0356 | **0.4570**±0.0498 | 0.7359±0.0219 |

Table 3: Results on SEED and HMC.

| Methods | Multi-task | SEED | | | HMC | | |
|---|---|---|---|---|---|---|---|
| | | Balanced Acc. | Cohen's Kappa | Weighted F1 | Balanced Acc. | Cohen's Kappa | Weighted F1 |
| SPaRCNet | ✗ | 0.5596±0.0244 | 0.3464±0.0372 | 0.5585±0.0297 | 0.4756±0.1109 | 0.3147±0.1315 | 0.4108±0.1310 |
| ContraWR | ✗ | 0.6106±0.0078 | 0.4220±0.0129 | 0.6137±0.0085 | 0.4242±0.0541 | 0.2340±0.0554 | 0.2987±0.0288 |
| CNN-Transformer | ✗ | 0.6161±0.0384 | 0.4262±0.0601 | 0.6150±0.0463 | 0.6573±0.0141 | 0.5961±0.0105 | 0.6896±0.0065 |
| FFCL | ✗ | 0.5808±0.0322 | 0.3732±0.0462 | 0.5743±0.0402 | 0.4427±0.0702 | 0.2542±0.0654 | 0.2902±0.0485 |
| ST-Transformer | ✗ | 0.5479±0.0091 | 0.3261±0.0169 | 0.5505±0.0091 | 0.2559±0.0141 | 0.0503±0.0183 | 0.1428±0.0122 |
| BIOT | ✗ | 0.7097±0.0024 | 0.5682±0.0051 | 0.7134±0.0027 | 0.6862±0.0041 | 0.6295±0.0113 | 0.7091±0.0147 |
| LaBraM-Base | ✗ | 0.7318±0.0019 | 0.5994±0.0031 | 0.7354±0.0021 | 0.7286±0.0101 | 0.6812±0.0073 | 0.7554±0.0024 |
| NeuroLM-B | ✓ | 0.5554±0.0075 | 0.3393±0.0117 | 0.5599±0.0068 | **0.6737**±0.0050 | **0.6188**±0.0057 | **0.7126**±0.0034 |
| NeuroLM-L | ✓ | 0.6006±0.0047 | 0.4067±0.0063 | 0.6048±0.0050 | 0.6658±0.0550 | 0.5929±0.0715 | 0.6896±0.0504 |
| NeuroLM-XL | ✓ | **0.6034**±0.0010 | **0.4082**±0.0036 | **0.6063**±0.0030 | 0.5761±0.1084 | 0.4795±0.1466 | 0.5883±0.1286 |

Table 4: Results on Workload and TUSL.

| Methods | Multi-task | Workload | | | TUSL | | |
|---|---|---|---|---|---|---|---|
| | | Balanced Acc. | AUC-PR | AUROC | Balanced Acc. | Cohen's Kappa | Weighted F1 |
| SPaRCNet | ✗ | 0.5977±0.0071 | 0.6638±0.0314 | 0.6717±0.0172 | 0.4185±0.0452 | 0.1399±0.0799 | 0.3500±0.0968 |
| ContraWR | ✗ | 0.6966±0.0332 | 0.7668±0.0408 | 0.7685±0.0317 | 0.5857±0.0662 | 0.3567±0.0968 | 0.5458±0.0798 |
| CNN-Transformer | ✗ | 0.5793±0.0230 | 0.5306±0.0459 | 0.5663±0.0349 | 0.3575±0.0151 | 0.0306±0.0179 | 0.2235±0.0251 |
| FFCL | ✗ | 0.7069±0.0197 | 0.7823±0.0099 | 0.7857±0.0234 | 0.3819±0.0688 | 0.0628±0.0888 | 0.2120±0.0786 |
| ST-Transformer | ✗ | 0.6103±0.0056 | 0.5716±0.0071 | 0.6375±0.0078 | 0.4000±0.0329 | 0.0860±0.0449 | 0.3793±0.0459 |
| BIOT | ✗ | 0.6655±0.0665 | 0.7189±0.0722 | 0.7342±0.0536 | 0.5758±0.0303 | 0.2012±0.0212 | 0.2394±0.0040 |
| LaBraM-Base | ✗ | 0.6609±0.0204 | 0.7174±0.0234 | 0.7272±0.0165 | 0.7625±0.0131 | 0.6407±0.0304 | 0.7614±0.0210 |
| NeuroLM-B | ✓ | 0.6172±0.0113 | 0.5824±0.0080 | **0.6253**±0.0160 | 0.6734±0.0436 | 0.5107±0.0617 | 0.6743±0.0394 |
| NeuroLM-L | ✓ | 0.6311±0.0250 | 0.5869±0.0155 | 0.6247±0.0339 | 0.5314±0.0530 | 0.2961±0.0810 | 0.5243±0.0680 |
| NeuroLM-XL | ✓ | **0.6345**±0.0442 | **0.5889**±0.0423 | 0.6130±0.0764 | **0.6845**±0.0304 | **0.5194**±0.0461 | **0.6839**±0.0297 |

HMC since sleep patterns are of low complexity and smaller models might be sufficient to capture the relevant features in EEG signals. On TUSL, the model performance appears to be not very stable due to the extremely limited data samples.

## 3.4 ABLATION ON ROBUSTNESS

Our instruction design for some datasets (TUEV, HMC, and TUSL) follows multiple-choice questions. To validate the robustness of NeuroLM, we enumerate the orders of options and randomly select one from all possible combinations during data fetching of the multi-task instruction tuning stage. Figure 5 illustrates the results on whether shuffling the options. We can conclude that on TUEV and HMC, NeuroLM with shuffle obtains comparable performance compared to those without shuffle. Nevertheless, it seems that the shuffle operation significantly degrades the performance on TUSL. We attribute this phenomenon to the lack of data for TUSL because TUSL has much fewer number of data samples compared to the other two datasets. It is expected that NeuroLM will achieve similar results if given more data. In general, NeuroLM has good robustness against arbitrary order of options, which indicates that NeuroLM does understand the linguistic meaning of the questions when predicting.

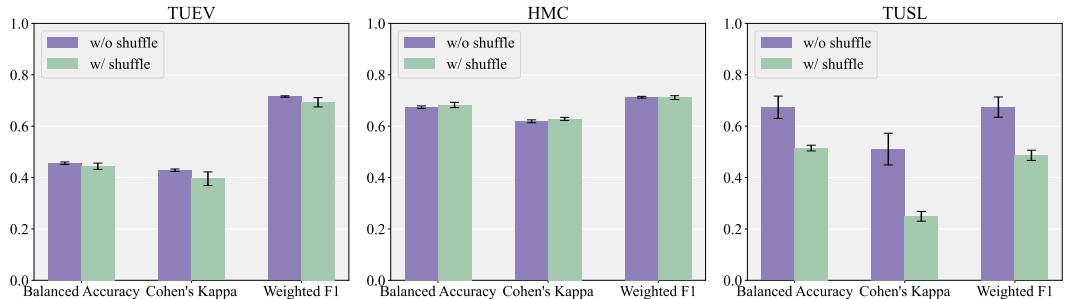

Figure 5: Ablation study on whether shuffling the options of instructions.

## 3.5 ABLATION ON INSTRUCTION DATA SIZE

We utilize TUAB, TUEV, and HMC datasets to scale the instruction data size and validate the performance of NeuroLM and other baseline methods, as these three datasets have a relatively large number of samples. The results, illustrated in Figure 6, show that NeuroLM demonstrates consistent performance under all conditions. For TUAB, NeuroLM, LaBraM, and CNN-Transformer exhibit stable performance. For TUEV and HMC, only NeuroLM and LaBraM are relatively unaffected by changes in data size. These findings indicate that NeuroLM is robust and maintains high performance even with varying instruction data sizes, highlighting its effectiveness in multi-task learning scenarios.

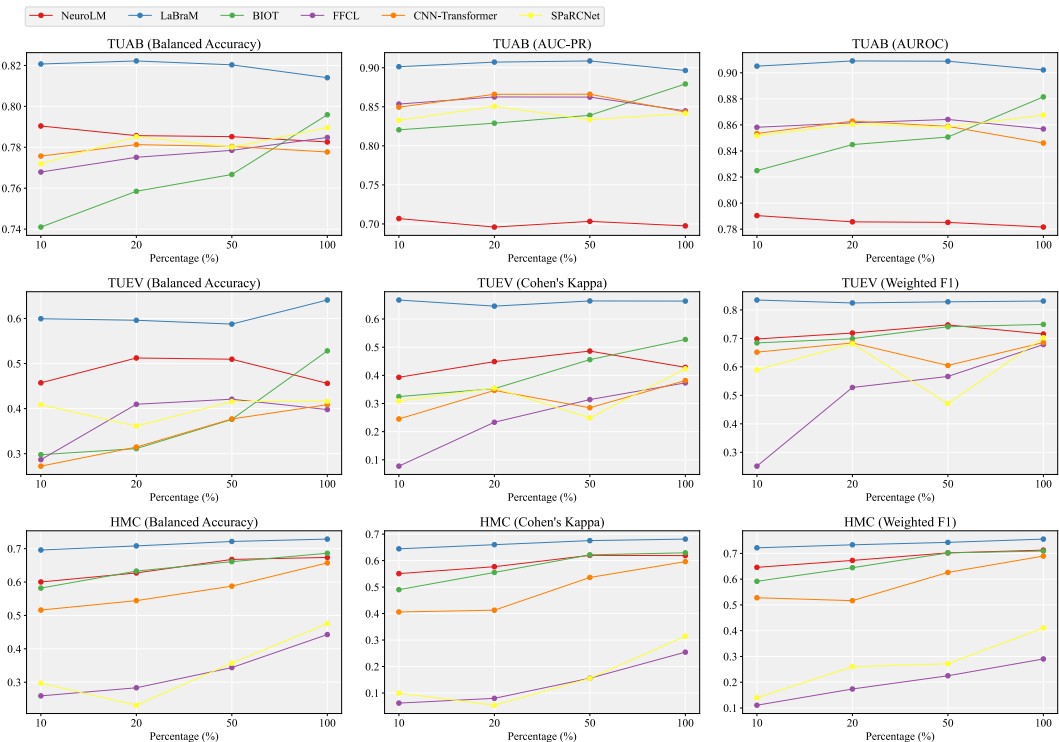

Figure 6: Comparison of different methods under different proportions of instruction data.

## 3.6 VISUALIZATION CURVES OF MULTI-CHANNEL AUTOREGRESSION

We visualize the pre-training loss, accuracy, and validation perplexity of NeuroLM in Figure 7. We observe that the loss stably converges while the validation perplexity decreases with training, which means NeuroLM can generalize well to unseen EEG data. Intuitively, a larger model with more parameters obtains smaller loss and perplexity. Additionally, NeuroLM-L achieves similar

validation perplexity with NeuroLM-XL, indicating that current pre-training data size still cannot satisfy the training with billion-level parameters.

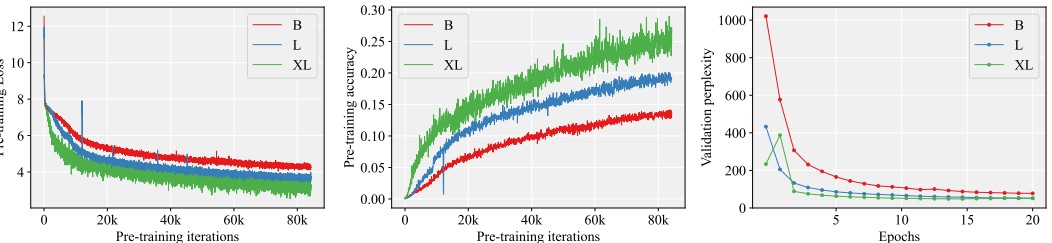

Figure 7: The training and validation visualization of multi-channel autoregressive pre-training.

## 3.7 Ablation on Multi-channel Autoregressive Pre-training

The proposed multi-channel autoregressive pre-training aims at mimicking current causal LLMs by predicting the next EEG tokens for each channel. It is expected to benefit downstream tasks through learning causal representations. We perform an ablation study to assess the impact of the proposed multi-channel autoregressive pre-training on NeuroLM. The results, shown in Figure 8, reveal a significant performance improvement when NeuroLM is pre-trained with this approach, underscoring the effectiveness of multi-channel autoregressive pre-training.

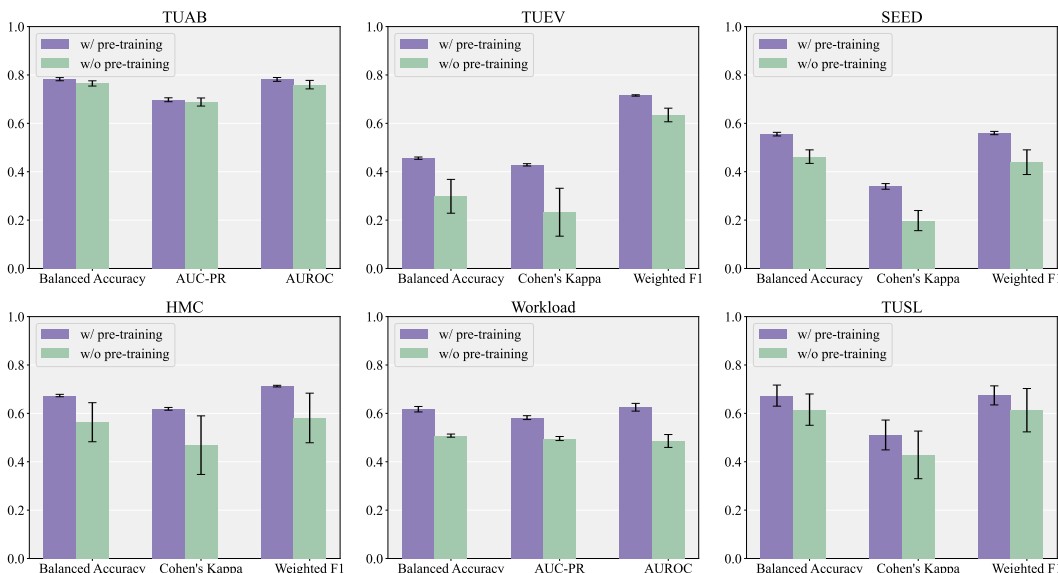

Figure 8: Ablation study on multi-channel autoregressive pre-training.

## 4 Conclusion

In this paper, we introduce NeuroLM, the first universal multi-task foundation model for EEG signal processing. By integrating EEG signals into a Large Language Model framework, NeuroLM leverages advanced text-aligned neural tokenizer embeddings, large-scale multi-channel autoregressive pre-training, and joint multi-task tuning to address the inherent challenges of EEG-based BCI and healthcare tasks. Our extensive experiments across six diverse EEG datasets demonstrate the model's superior performance in multi-task learning and inference. Overall, NeuroLM represents a significant step forward in the field of brain-computer interfaces and healthcare domains, showcasing the great potential of LLMs to revolutionize EEG signal processing and multi-task learning. We believe that NeuroLM will pave the way for more sophisticated and versatile EEG applications, ultimately enhancing the interaction between humans and machines.

ACKNOWLEDGMENTS

B. L. Lu acknowledges the following grants: STI 2030-Major Projects+2022ZD0208500, National Natural Science Foundation of China (Grant No. 62376158), Shanghai Municipal Science and Technology Major Project (Grant No. 2021SHZD ZX), Medical-Engineering Interdisciplinary Research Foundation of Shanghai Jiao Tong University "Jiao Tong Star" Program (YG2023ZD25, YG2024ZD25), Shanghai Pilot Program for Basic Research - Shanghai Jiao Tong University (No. 21TQ1400203), and Shanghai Jiao Tong University SEIEE-Shanghai EmoRays Technology Co., Ltd Joint Laboratory of Affective Brain-Computer Interfaces.

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

## A  RELATED WORK

**Large-scale Pre-training for Neural Signals.** With the success of self-supervised learning in computer vision and natural language processing, several studies have emerged to learn effective EEG representations. Kostas et al. (2021) first propose BENDER, which adapts contrastive learning to derive compressed representations from massive EEG datasets. MMM (Yi et al., 2023) introduces a pre-training framework with multi-dimensional position encoding, multi-level channel hierarchy, and a multi-stage pre-training strategy to learn topology-agnostic representations. BIOT (Yang et al., 2023a) tokenizes diverse biosignals into unified segments, enabling cross-data learning despite mismatched channels, variable lengths, and missing values. Brant (Zhang et al., 2023) pre-trains on a large corpus of private intracranial EEG data, capturing long-term dependencies, spatial correlations, and both time and frequency domains. Following BIOT, LaBraM (Jiang et al., 2024) further leverages large-scale 2,500 hours public EEG data, and innovatively introduces a neural tokenizer that encodes continuous EEG signals into discrete codes for masked EEG modeling, thus obtaining a considerable improvement. Unfortunately, all these methods require fine-tuning for specific downstream tasks and cannot perform multi-task learning and inference.

**Multimodal Large Language Models.** Recent years have seen remarkable achievements of LLMs. In light of the complementarity between language and other modalities, Multimodal Large Language Models have been a rising hotspot. The release of GPT-4 (Achiam et al., 2023) shows the extraordinary multimodal understanding and generation abilities, thus leading to a research frenzy over MLLMs. LLaVA (Liu et al., 2023) connects a vision encoder and an LLM, introducing the idea of visual instruction tuning for general-purpose visual and language understanding. Similarly, Zhu et al. (2023) propose MiniGPT-4, which aligns a frozen visual encoder with a frozen advanced LLM, presenting numerous advanced multi-modal abilities. Different from the above MLLMs, CogVLM (Wang et al., 2023) bridges the gap between the frozen pre-trained LLM and visual encoder by a trainable visual expert in the attention and FFN layers. Chen et al. (2024) present InternVL-1.5, closing the capability gap between open-source and proprietary commercial MLLMs by utilizing a strong vision encoder, dynamic high-resolution, and high-quality bilingual dataset.

## B  INSTRUCTION DESIGN

Table 5: Information of instruction design for downstream datasets.

| Dataset | Instruction Description |
|---|---|
| TUAB | [SEP] Question: Is this EEG segment abnormal? Answer: {Yes, No} [END] |
| TUEV | [SEP] Question: Which event type does this EEG segment belong to? Options: (A) spike and slow wave. (B) generalized periodic epileptiform discharge. (C) periodic lateralized epileptiform discharge. (D) eye movement. (E) artifact. (F) background. Answer: {(A), (B), (C), (D), (E), (F)} [END] |
| SEED | [SEP] Question: Which emotion type does this EEG segment belong to? Answer: {Positive, Neutral, Negative} [END] |
| HMC | [SEP] Question: Which sleep type does this EEG segment belong to? Options: (A) Wake. (B) NREM-1. (C) NREM-2. (D) NREM-3. (E) REM. Answer: {(A), (B), (C), (D), (E)} [END] |
| Workload | [SEP] Question: Is this EEG segment of high workload? Answer: {Yes, No} [END] |
| TUSL | [SEP] Question: Which type does this EEG segment belong to? Options: (A) background. (B) seizure. (C) slowing. Answer: {(A), (B), (C)} [END] |

## C  PRE-TRAINING DATASET DESCRIPTION

We utilize multiple EEG datasets with various configurations. The detail information of all the datasets are listed in 6. The total time after data cleaning is close to 25,000 hours.

Table 6: Information of datasets used for pre-training.

| Dataset | #Channel | Rate (Hz) | Time (h) | Description |
|---|---|---|---|---|
| TUEG (Obeid & Picone, 2016) | 17-23 | 250-1024 | ∼24,000 | A corpus of 26,846 clinical EEG recordings collected at Temple University Hospital. |
| SEED Series (Zheng et al., 2018; Liu et al., 2021; 2022) | 62 | 1000 | 170.54 | These datasets including SEED-IV (15 subjects), SEED-V (20 subjects), SEED-GER (8 subjects), and SEED-FRA (8 subjects) in response to emotional videos. |
| BCI Competition IV-1 (Blankertz et al., 2007) | 59 | 1000 | 8.21 | A motor imagery dataset containing 2 classes of left hand, right hand, foot (+ idle state) for 7 subjects. |
| Emobrain (Savran et al., 2006) | 64 | 1024 | 4.94 | A multimodal emotion dataset including 16 subjects. The emotions were elicited through a selected subset of the IAPS dataset. |
| Grasp and Lift (Luciw et al., 2014) | 32 | 500 | 11.72 | A dataset containing 12 subjects performing grasp-and-lift (GAL) trials. |
| Inria BCI (Margaux et al., 2012) | 56 | 600 | 29.98 | A P300-based spelling dataset including 26 subjects. |
| Motor Movement/Imagery (Schalk et al., 2004) | 64 | 160 | 47.3 | A motor imagery dataset consisting of 109 volunteers performing 2 baseline tasks, motor movement, and motor imagery. |
| Raw EEG Data (Trujillo, 2020) | 64 | 256 | 34.35 | EEG was recorded during reported Information-Integration categorization and reported multidimensional Rule-Based categorization tasks. |
| Resting State (Trujillo et al., 2017) | 64 | 256 | 3.04 | A dataset comprising 22 subjects for a resting task of 8 mins with 4 mins of eyes closed and 4 mins of eyes open. |
| Siena Scalp EEG Database (Detti et al., 2020) | 31 | 512 | 30.47 | A database consisting of 14 patients. |
| SPIS Resting State (Torkamani-Azar et al., 2020) | 64 | 2048 | 0.83 | A dataset including 10 subjects, 2.5 minutes recording in eyes-closed and eyes-open prior to a 105-minute session of Sustained Attention to Response Task with fixed-sequence and varying ISIs. |
| Target Versus Non-Target (Korczowski et al., 2019) | 32 | 512 | 16 | A dataset including 50 subjects playing Brain Invaders, a visual P300 Brain-Computer Interface using oddball paradigm with adaptive Riemannian Geometry (no-calibration). |
| Self-collected EEG corpus (Jiang et al., 2023; 2021; Luo et al., 2022; Li et al., 2021; Tao & Lu, 2020) | 62 | 1000 | 342.23 | A mixed self-collected EEG datasets of more than 140 subjects under various conditions. |

# D DETAILED EXPERIMENTAL SETTINGS

## D.1 HYPERPARAMETER SETTINGS

Table 7: Hyperparameters for neural tokenizer.

| Hyperparameters | | Values |
|---|---|---|
| Temporal Encoder | Iput channels | {1,16,16} |
| | Output channels | {16,16,16} |
| | Kernel size | {15,3,3} |
| | Stride | {8,1,1} |
| | Padding | {7,1,1} |
| Transformer encoder layers | | 12 |
| Transformer decoder layers | | 3 |
| Hidden size | | 768 |
| MLP size | | 3072 |
| Attention head number | | 12 |
| Codebook size | | 8192×128 |
| EEG Batch size | | 512 |
| Text Batch size | | 128 |
| Peak learning rate | | 5e-5 |
| Minimal learning rate | | 1e-5 |
| Learning rate scheduler | | Cosine |
| Optimizer | | AdamW |
| Adam $\beta$ | | (0.9,0.999) |
| Weight decay | | 1e-4 |
| Total epochs | | 50 |
| Warmup epochs | | 5 |
| Data overlap | | None |
| Gradient clipping | | None |

Table 8: Hyperparameters for autoregressive pre-training.

| Hyperparameters | NeuroLM-B | NeuroLM-L | NeuroLM-XL |
|---|---|---|---|
| Model size | 254M | 500M | 1696M |
| Transformer encoder layers | 12 | 24 | 48 |
| Hidden size | 768 | 1024 | 1600 |
| MLP size | 3072 | 4096 | 6400 |
| Attention head number | 12 | 16 | 25 |
| EEG batch size | | 480 (B), 512 (L, XL) | |
| Text batch size | | 32 (B), 64 (L, XL) | |
| Peak learning rate | | 6e-4 | |
| Minimal learning rate | | 6e-5 | |
| Learning rate scheduler | | Cosine | |
| Optimizer | | AdamW | |
| Adam $\beta$ | | (0.9,0.95) | |
| Weight decay | | 0.1 | |
| Total epochs | | 20 | |
| Warmup epochs | | 2 | |
| Data overlap | | None | |
| Gradient clipping | | 1 | |

Table 9: Hyperparameters for instruction tuning.

| Hyperparameters | Values |
|---|---|
| Instruction batch size | 512 |
| Text batch size | 128 |
| Peak learning rate | 5e-4 (B), 5e-5 (L), 2e-5 (XL) |
| Minimal learning rate | 5e-5 (B), 5e-6 (L), 2e-6 (XL) |
| Learning rate scheduler | Cosine |
| Optimizer | AdamW |
| Adam $\beta$ | (0.9,0.95) |
| Weight decay | 0.1 |
| Total epochs | 5 (B, L), 3 (XL) |
| Warmup ratio | 0.1 |
| Gradient clipping | 1 |

## D.2 METRICS

Considering the class imbalance of most downstream EEG datasets, we use the following metrics for comparison:

- **Balanced Accuracy**: The average of recall (sensitivity) obtained on each class. It is particularly useful for evaluating classification performance on imbalanced datasets. This metric is particularly useful when evaluating models on imbalanced datasets.

- **AUC-PR**: A performance measurement for binary classification problems. It is the area under the curve plotted with precision (y-axis) against recall (x-axis) for different threshold values.

- **AUROC**: It is the area under the curve plotted with the true positive rate (sensitivity) on the y-axis and the false positive rate (1 - specificity) on the x-axis for different threshold values. AUROC provides an aggregate measure of performance across all possible classification thresholds, indicating the ability of the model to distinguish between classes.

- **Cohen's Kappa**: A measure of agreement between categorical variables $X$ and $Y$, calculated from the observed and expected frequencies on the diagonal of a square contingency table. It is used for multi-class classification.

- **Weighted F1**: The weighted F1 score is the harmonic mean of precision and recall, taking into account the support (the number of true instances) of each class. The weighted F1 score accounts for class imbalance by giving more importance to classes with a higher number of instances.

AUROC and Cohen's Kappa are used as the monitor score for binary classification and multi-class classification, respectively.

## D.3 BASELINES

We mainly consider BIOT (Yang et al., 2023a) and the state-of-the-art EEG foundation model LaBraM (Jiang et al., 2024) as our baseline method, where BIOT is a generic biosignal learning model pre-trained on multiple datasets in a supervised way, and LaBraM is pre-trained on 2,500 hours data through masked EEG modeling and has learned generic representations for various EEG signals. Five other supervised methods including SPaRCNet (Jing et al., 2023), ContraWR (Yang et al., 2023b), CNN-Transformer (Peh et al., 2022), FFCL (Li et al., 2022), and ST-Transformer (Song et al., 2021) are also utilized as our baselines. As there are no multi-task methods available in EEG signal processing yet, these baselines are solely fine-tuned on each downstream dataset and cannot perform multiple tasks. We use the default settings for these baselines in the BIOT paper. The batch size is 512 for TUAB, TUEV, SEED, and HMC. As the data size of Workload and TUSL is particularly small, the batch size of these two datasets is set to 32 and 16, respectively.

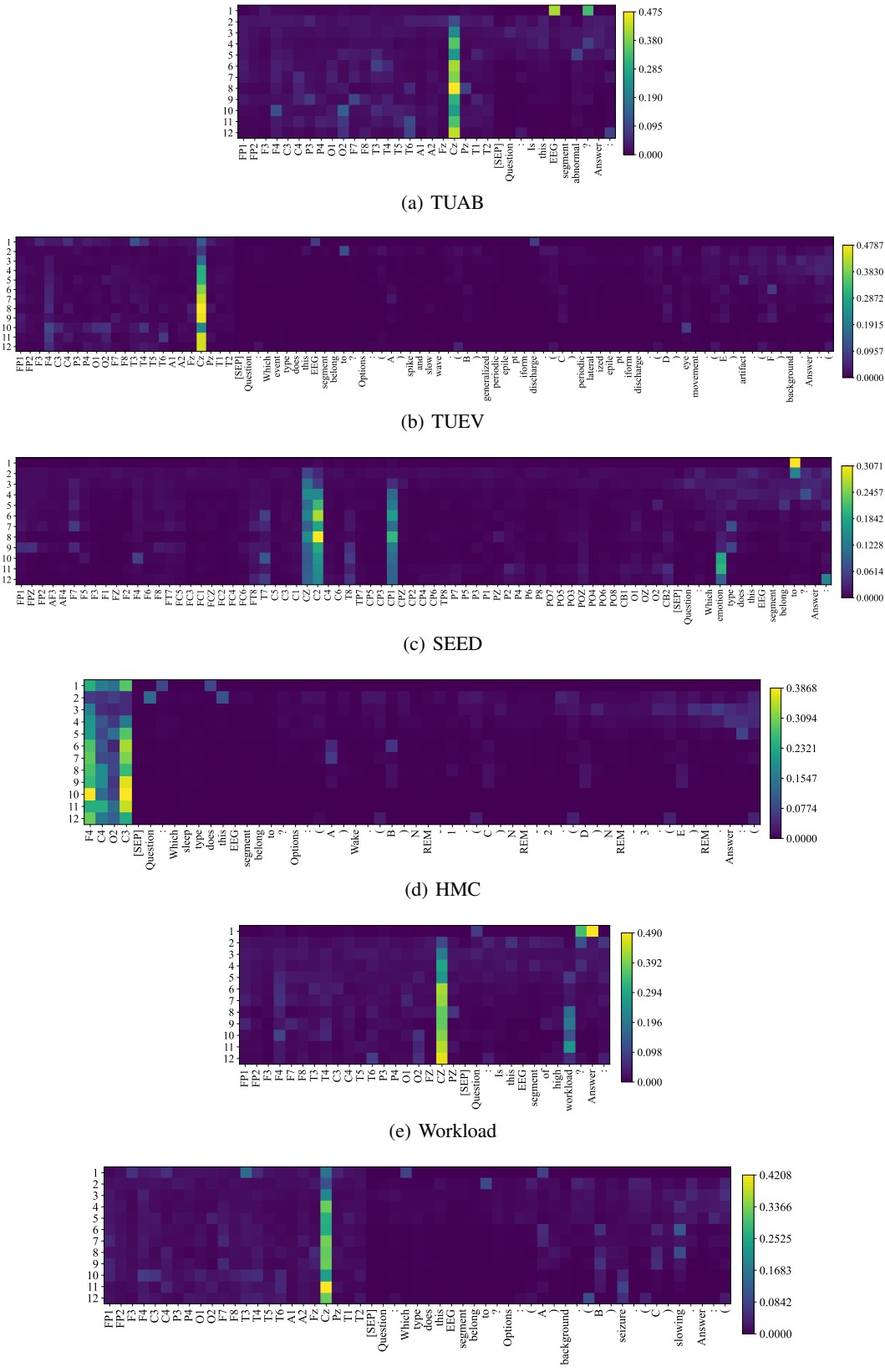

Figure 9: The attention value on other datasets. The vertical axis denotes the Transformer layers.

# E ATTENTION VISUALIZATION

To explore the mechanism of NeuroLM, we visualize the attention scores of the answer parts in the instructions for all 12 Transformer layers, as drawn in Figure 9. Firstly, we observed several commonalities across datasets: For the text part, the attention tends to concentrate more in shallow layers whereas for the EEG part, attention gains more in deeper layers. This pattern suggests that NeuroLM primarily processes text questions in the shallow layers and focuses on EEG tokens in the deeper layers to generate answers. Interestingly, in the case of multiple-choice questions, NeuroLM pays close attention to the options (A, B, C, etc.) between the 6th and 9th layers. Analyzing critical EEG channels for different tasks, we find that NeuroLM seems to aggregate information to Cz for most datasets. For HMC, F4 and C3 are crucial, while O2 is less effective in sleep stage classification.

# F ANALYSIS OF NEURAL TOKENIZER

## F.1 ABLATION ON TEMPORAL-FREQUENCY PREDICTION

Temporal and frequency domains are two pivotal aspects of EEG signals. To investigate the importance of these two domains for different downstream tasks, we study three variants by setting the reconstruction target in neural tokenizer training as only the temporal domain, only the frequency domain, and both temporal and frequency domains (original NeuroLM). Figure 10 shows the comparison between the three variants. Interestingly, it can be found that the temporal domain plays a more crucial role on TUAB, Workload, and TUSL. On the contrary, reconstructing the frequency components obtains better performance on TUEV, SEED, and HMC, indicating that the frequency domain is of great importance for event classification, emotion recognition, and sleep stage classification. By combining the two domains, most tasks achieve similar or higher performance, demonstrating the effectiveness of our neural tokenizer that excavates compact EEG representations for language models.

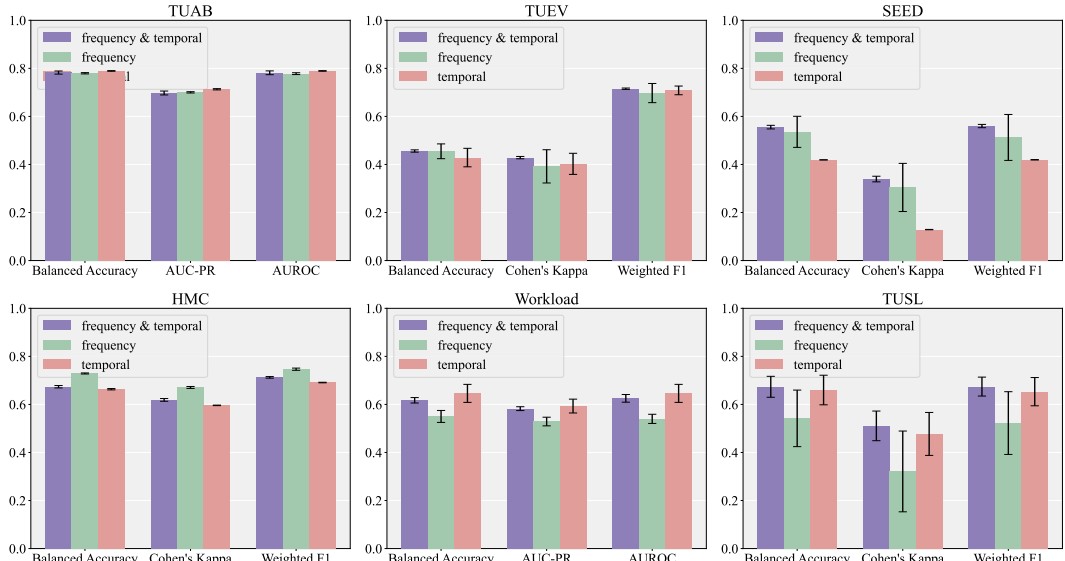

Figure 10: Ablation study on reconstructing temporal or frequency domain in neural tokenizer.

## F.2 VISUALIZATION OF EEG AND TEXT EMBEDDINGS

To evaluate the effectiveness of EEG-text embedding space alignment, we visualize the embeddings in Figure 11 using t-SNE (Van der Maaten & Hinton, 2008). The EEG embeddings expand outside the text space without alignment. In this case, we find that the model fails to predict the answers we expect in multi-task instruction tuning, i.e., the model will output random words instead of

options like (A), (B), or (c) in choice questions, leading to near-zero scores across most metrics on downstream tasks. This outcome appears to stem from disordered attention scores, as EEG and text embeddings remain in separate spaces. When training with alignment, the EEG space mostly aligns with text space, resulting in normal prediction in instruction tuning, proving the necessity of EEG-text alignment.

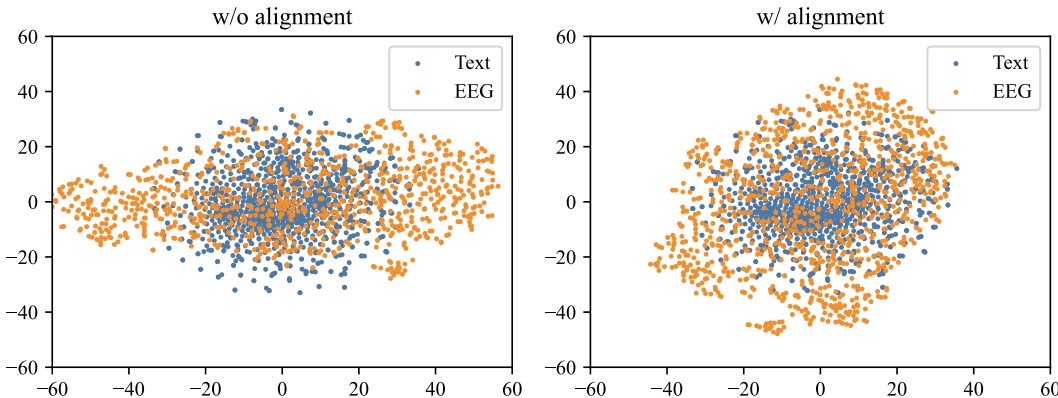

Figure 11: Representation visualization of EEG and text by t-SNE. **Left**: training neural tokenizer without alignment. **Right**: training neural tokenizer with alignment.

## G   ABLATION ON DIFFERENT PRE-TRAINING EPOCHS

We conduct an ablation study on tuning the pre-trained models from different epochs to testify the best pre-training epoch. As shown in Table 10 11 12, we use the pre-trained models of 5, 10, 15, and 20 epochs. Bold represents the best results and underline represents the second best results. It can be found that pre-training for 20 epochs obtains the most bold and underlined results. The performance of 5 epochs gets its best on SEED and HMC while the model from 10 epochs achieves the best result on Workload. Overall, pre-training for more epochs can lead to good performance in different tasks.

Table 10: Results on TUAB and TUEV.

| Pre-trained Epochs | TUAB | | | TUEV | | |
|---|---|---|---|---|---|---|
| | Balanced Acc. | AUC-PR | AUROC | Balanced Acc. | Cohen's Kappa | Weighted F1 |
| 5 | 0.7763±0.0037 | 0.6891±0.0040 | 0.7762±0.0038 | 0.4609±0.0696 | 0.3914±0.0670 | 0.6975±0.0249 |
| 10 | 0.7738±0.0040 | **0.6995**±0.0038 | 0.7647±0.0143 | **0.4693**±0.0175 | **0.4625**±0.0091 | **0.7353**±0.038 |
| 15 | 0.7780±0.0050 | 0.6951±0.0054 | 0.7735±0.0112 | 0.4557±0.0277 | 0.4216±0.0215 | 0.7131±0.0101 |
| 20 | **0.7826**±0.0065 | 0.6975±0.0081 | **0.7816**±0.0079 | 0.4560±0.0048 | 0.4285±0.0048 | 0.7153±0.0028 |

Table 11: Results on SEED and HMC.

| Pre-trained Epochs | SEED | | | HMC | | |
|---|---|---|---|---|---|---|
| | Balanced Acc. | Cohen's Kappa | Weighted F1 | Balanced Acc. | Cohen's Kappa | Weighted F1 |
| 5 | **0.5641**±0.0103 | **0.3505**±0.0172 | **0.5679**±0.0123 | **0.6956**±0.0136 | **0.6269**±0.0053 | 0.7118±0.0048 |
| 10 | 0.5553±0.0089 | 0.3376±0.0143 | 0.5592±0.0091 | 0.6763±0.0054 | 0.6161±0.0069 | 0.7065±0.0057 |
| 15 | 0.5543±0.0156 | 0.3365±0.0244 | 0.5572±0.0164 | 0.6543±0.0151 | 0.5991±0.0153 | 0.6674±0.0265 |
| 20 | 0.5554±0.0075 | 0.3393±0.0117 | 0.5599±0.0068 | 0.6737±0.0050 | 0.6188±0.0057 | **0.7126**±0.0034 |

Table 12: Results on Workload and TUSL.

| Pre-trained Epochs | Workload | | | TUSL | | |
|---|---|---|---|---|---|---|
| | Balanced Acc. | AUC-PR | AUROC | Balanced Acc. | Cohen's Kappa | Weighted F1 |
| 5 | 0.5816±0.0235 | 0.5483±0.0160 | 0.5815±0.0236 | 0.5342±0.0235 | 0.2950±0.0345 | 0.5241±0.0260 |
| 10 | **0.6540**±0.0192 | **0.6123**±0.0165 | **0.6501**±0.0178 | 0.5920±0.0560 | 0.3884±0.0876 | 0.5984±0.0574 |
| 15 | 0.5701±0.0282 | 0.5426±0.0177 | 0.5682±0.0255 | 0.5910±0.0629 | 0.3915±0.0999 | 0.5868±0.0536 |
| 20 | 0.6172±0.0113 | 0.5824±0.0080 | 0.6253±0.0160 | **0.6734**±0.0436 | **0.5107**±0.0617 | **0.6743**±0.0394 |

## H    DISCUSSION

**Limitations.** NeuroLM represents the first attempt to integrate various EEG downstream tasks into a unified model, achieving promising results across multiple downstream datasets. However, it has some limitations: 1) Although NeuroLM can surpass certain single-task baselines, it still lags behind state-of-the-art methods that are end-to-end trained on each downstream dataset. 2) NeuroLM is somewhat sensitive to hyperparameter settings, and may not yield satisfactory results without careful tuning. 3) With limited high-quality EEG-text pairs available, this paper only employs coarse-grained alignment between EEG and language, i.e., space-wise alignment, which can pose challenges for LLMs in extracting useful information from EEG tokens.

**Outlook.** Reflecting on the outlook part of the LaBraM paper, this work explores the first and third suggested directions. Looking ahead, we foresee several potential improvements: 1) Utilizing more advanced LLMs as the base models. While this paper uses GPT-2, a relatively small LLM, and still achieves promising results in the multi-task paradigm, leveraging newer, more advanced open-source LLMs such as LLaMA 3 (Dubey et al., 2024) may significantly enhance NeuroLM's multi-task learning capabilities. 2) Adopting the mixture-of-experts approach is another promising direction. Given the modality gap between EEG and language, using modality-specific experts may improve multimodal learning with LLMs. 3) Developing finer-grained EEG and text alignment methods, such as describing EEG samples with predefined sentences and aligning EEG and text descriptions at the VQ training stage by adding a contrastive learning loss, may further enhance performance.

