# OpenReview forum: "NeuroLM: A Universal Multi-task Foundation Model for Bridging the Gap between Language and EEG Signals"
_ICLR.cc/2025/Conference — ICLR 2025 Poster_

### Official Review · Reviewer_gQYU · 2024-10-25

**Soundness:** 3
**Presentation:** 2
**Contribution:** 3
**Rating:** 6
**Confidence:** 4

**Summary:**

The paper introduces NeuroLM, a universal multi-task foundation model for EEG signal processing that consists of three key stages: a text-aligned neural tokenizer, multi-channel autoregressive pre-training, and multi-task instruction tuning. Traditional EEG pre-trained models often require full fine-tuning for each downstream task, resulting in resource inefficiencies. NeuroLM addresses this issue by leveraging Large Language Models (LLMs) and treating EEG signals as a foreign language, demonstrating the potential of this multi-task learning paradigm across multiple EEG-related tasks.

**Strengths:**

- The approach of treating EEG signals as a form of language and integrating them with LLMs is novel and creative, offering a new direction for EEG signal processing.

- NeuroLM shows potential in unifying different EEG tasks within a single model, reducing the need for task-specific fine-tuning and streamlining BCI applications.

- The experiments cover six diverse datasets, demonstrating the model's versatility and robustness in multi-task settings.

**Weaknesses:**

My concerns mainly focus on four areas, which are detailed in the Questions section. These include the effectiveness of EEG-Text Embedding Space Alignment, the theoretical foundation of multi-channel autoregressive pre-training, the lack of discussion on instruction design's impact on downstream tasks, and the absence of an analysis on data curriculum or input order. I hope the authors can address these concerns in the rebuttal stage.

**Questions:**

Concern 1:
The paper introduces an EEG-Text Embedding Space Alignment module in Figure 2, built on LaBraM’s neural tokenizer. However, I have concerns regarding the effectiveness and explanation of this alignment.

Fundamentally, EEG and Text have no inherent connection, especially since the EEG signals are not language-evoked. The use of a single gradient reverse layer may only distinguish EEG from text domains rather than fully align their embeddings. To clarify the module's effectiveness, could the authors:

1) Provide visualizations, such as t-SNE plots, comparing EEG and Text embedding spaces before and after alignment to illustrate the degree of alignment achieved.
2) Clarify the source of the Text embeddings and their generation method. The authors do not explain where these embeddings come from or how they are generated, leaving an important gap in understanding.
3) Include an ablation study to quantify the impact of the EEG-Text Embedding Space Alignment module on downstream task performance, showing its necessity and contribution to the overall model.

Concern 2:
The Theory Analysis section in 2.2 of Multi-channel Autoregressive Pre-training provides a high-level explanation by drawing an analogy to VAE, optimizing for reconstruction loss and KL divergence, but it lacks a detailed explanation of how causal relationships between EEG channels are learned or how this approach is theoretically grounded in the autoregressive framework.

To enhance the theoretical foundation of the multi-channel autoregressive approach, could the authors:

Include mathematical formulations demonstrating how this approach models dependencies between EEG channels and captures causal relationships.

Concern 3:
The paper lacks a detailed discussion regarding instruction design and how variations in prompt formulations affect downstream task performance. While instruction tuning addresses multiple tasks, the authors do not examine how different prompts might affect task performance. Could the authors provide an analysis of how different prompts influence task performance?

Concern 4:
The paper does not sufficiently discuss the impact of data curriculum or input order on the results. Although instruction tuning is introduced, there is no exploration of whether sequencing the data in a specific way (e.g., starting with simpler tasks or data) influences model performance across the various downstream tasks. It would be helpful if the authors could provide an analysis of how the ordering of data inputs affects model learning and task outcomes.

---

> ### Author Response · Authors · 2024-11-19
> **Response**
>
> Thanks for your wisdom and valuable comments. We sincerely appreciate your recognition of our work. Here are our responses to your questions and concerns point-by-point.
> > **[Q1]** The effectiveness of EEG-Text Embedding Space Alignment
> 1. The visualizations of EEG and text embeddings are presented in Appendix F.2, which you may have missed. Without alignment, the EEG embeddings expand outside the text space, but with alignment training, the EEG embeddings are brought closer to the text space, achieving partial overlap.
> 2. We apologize for not clarifying the source of the text embeddings. For alignment, we use embeddings from GPT-2’s text vocabulary, and each batch is randomly sampled from the vocabulary. We have added this information in Section 3.2.
> 3. In our ablation study, we found that without EEG-text embedding alignment, the model fails to produce meaningful predictions in multi-task instruction tuning. Specifically, it outputs random words instead of the expected options (e.g., (A), (B), or \(C) in choice questions), resulting in near-zero scores across most downstream task metrics. This misalignment likely stems from disordered attention scores, as EEG and text embeddings remain in separate spaces. We discussed this phenomenon in Appendix F.2.
>
> > **[Q2]** The theoretical foundation of multi-channel autoregressive pre-training
> - Thank you for this insightful question. We agree that further theoretical grounding could enhance the understanding of how NeuroLM models dependencies between EEG channels and captures causal relationships in an autoregressive framework. In response, we have modified the Theory Analysis section in 2.2 to provide a more detailed formulation:
> $-\sum_{i=1}^N(E_{z_i\sim q_\phi(z|x_i)}[-\log p_\psi(y_i|z_i)]-\sum_{j=2}^T\log p_\theta(z_{i,j}|x_{i,<j}))$.
> Specifically, we revised the second term in the Equation (7) to explicitly reflect causal relationships by using a negative log-likelihood loss that accounts for latent variables across all channels at each time step. $z_{i,j}$ denotes latent variables of the i-th sample of all channels at time step j and $z_i=[z_{i,1},...,z_{i,T}]$, while $x_{i,<j}$ denotes the input of the i-th sample of all channels prior to time step j. This adjustment clarifies how the model learns dependencies over time and across channels, capturing the causal relationships inherent in EEG data. By conditioning each latent variable on past time steps, our approach allows NeuroLM to model multi-channel autoregressive relationships, which are crucial for capturing sequential dependencies in EEG signals. We hope this revision strengthens the theoretical foundation of our multi-channel autoregressive approach and provides a clearer mathematical basis for how NeuroLM models EEG data dependencies.
>
> > **[Q3]** The lack of discussion on instruction design's impact on downstream tasks
> - Thank you for this insightful question. We agree that the design of instructions and variations in prompt formulations may affect task performance. In this work, we incorporated some prompt-design tricks, such as randomizing the order of options in multi-choice tasks, as discussed in Section 3.4. This helped us ensure that NeuroLM could generalize to various option orders, thus increasing robustness across tasks.
> We also considered more complex prompt variations, such as introducing a chain-of-thought approach for EEG tasks. However, applying such advanced prompting techniques to EEG data is currently challenging due to the inherent difference in processing language and neural signals. Developing effective ways to incorporate these techniques would add significant complexity, and we see this as a promising direction for future research.
>
> > **[Q4]** The absence of an analysis on data curriculum or input order.
> - Thank you for this thoughtful question. In this study, we applied the simplest approach to data ordering by randomly shuffling the inputs across tasks. Our results show that even with this straightforward strategy, NeuroLM achieves promising performance across diverse datasets with varied sizes and complexities. This suggests that NeuroLM is robust to input order, but we agree that a structured data curriculum could further optimize performance. Exploring data sequencing strategies, such as introducing tasks in increasing complexity, could be a valuable avenue for future work. Given the scope of this study, however, including and exploring this intriguing idea would add excessive and overly varied content to the paper. Curriculum learning is indeed an exciting and worthwhile direction for future research, as it may offer additional performance gains beyond our initial findings.

---

> > ### Comment · Reviewer_gQYU · 2024-11-24
> > **Official Comment by Reviewer gQYU**
> >
> > Thanks for the authors' detailed reply and clarifications.
> >
> > Regarding Q1, as the authors mentioned, the EEG-Text Embedding Space Alignment module plays a crucial role in overcoming the issue of generating meaningless predictions during fine-tuning. This underscores the importance of the module. At the same time, it also highlights the significant semantic gap when forcibly integrating EEG signals into LLMs, emphasizing the necessity and challenges of aligning embeddings in such a framework.
> >
> > Although Q3-Q4 lack more quantitative results or in-depth analysis, the explanations provided have made the points clearer.
> >
> > However, I still believe that there is room for improvement in certain aspects, which is why I have decided to maintain my current score.
> >
> > Overall, I acknowledge the contribution of this paper. I believe NeuroLM has made a meaningful impact in this field. As the authors pointed out, this work introduces a novel perspective on EEG signal processing and demonstrates the potential of integrating EEG with LLMs.

---

> > > ### Author Response · Authors · 2024-11-24
> > >
> > > Thank you for your recognition of our paper! We are deeply grateful for your review, which has greatly assisted us in supplementing and perfecting our paper.

---

### Official Review · Reviewer_hyNm · 2024-10-25

**Soundness:** 2
**Presentation:** 2
**Contribution:** 2
**Rating:** 6
**Confidence:** 3

**Summary:**

Introduction of **NeuroLM**: a multi-task foundational model to process EEG signals. It leverages the capabilities of LLMs, employing a unified approach encoding the signals into discrete tokens. It addresses challenges like the usual need for individual task fine-tuning.
Model is utilizing a text-aligned neural tokenizer trained using an adversarial strategy and multi-channel autoregressive pre-training to enhance the understanding of the signals. Experimental results demonstrate a competitive performance across 6 different tasks.

**Strengths:**

- Innovative Idea : build the first EEG multi-task foundation model.
- Trained on large-scale EEG dataset (> 25K hours), diverse data sources enhance model's generalization ability.
- Consider EEG signals as foreign language and expand scope of LLMs including EEG tasks
- Instruction Tuning for Multi-Tasking: joint multi-task instruction handle various tasks in a unified manner.

**Weaknesses:**

- Absence of Fine-Tuning Examples: The authors position NeuroLM as a foundation model and present solid performance across six downstream tasks. However, the study does not include an exploration of fine-tuning capabilities to demonstrate NeuroLM's potential for achieving SOTA performance which could underscore its utility in practical applications.

- Dependency on GPT-2: The authors only test GPT-2 so this study limits insights into the broader potential of this innovative approach. The model aims to bridge EEG and language so evaluating additional language models is necessary for a more comprehensive understanding of the model robustness and adaptability.

- EEG-Text Alignment: The use of adversarial training is innovative. The absence of ablation studies limits understanding of its specific impact. It could enhance paper's clarity and support the results if it was shown how this approach contributes to the model's performance.

- Complexity and Reproducibility: The pipeline's complexity may pose challenges for reproducibility. Enhanced clarity in certain sections would better support the community in replicating and extending this work.

**Questions:**

- How robust is NeuroLM across different LLM architectures?
- What role does adversarial training play compared to alternative strategies?
- Can you show some examples of fine-tuning that lead to SOTA performance?

---

> ### Author Response · Authors · 2024-11-19
> **Response**
>
> Thanks for your wisdom and valuable comments. Here are our responses to your questions and concerns point-by-point.
> > **[W1 & Q3]** Absence of Fine-Tuning Examples
> - The primary contribution of NeuroLM lies in its unified multi-task instruction-tuning approach, which differentiates it from fine-tuning-focused models like LaBraM. Our goal was to establish a framework that generalizes across diverse EEG tasks without task-specific fine-tuning, showcasing NeuroLM’s flexibility and potential for handling new tasks or prompts directly through instruction tuning.
> That said, we recognize that exploring fine-tuning capabilities could potentially enhance NeuroLM’s task-specific performance and further demonstrate its utility in practical applications. Achieving this would likely involve several future improvements: 1) **Stronger Language Models**: NeuroLM currently utilizes GPT-2, a relatively small and older LLM. Incorporating more advanced LLMs, such as LLaMA 3 series could significantly improve both the model's generalization capabilities and fine-tuned task-specific performance. 2) **Fine-Grained Contrastive Learning-Based Alignment**: Enhancing the EEG-text alignment process through contrastive learning could enable the model to better align EEG signals with textual descriptions, leading to more robust representations and performance gains. 3) **Expanding Data Resources**: Pretraining on a larger and more diverse dataset of EEG signals could improve the model’s capacity to generalize across tasks and handle fine-tuning more effectively. Additionally, using high-quality annotated datasets could further refine task-specific outcomes.
> These directions represent opportunities for future work that can build on NeuroLM’s foundation. In this paper, our primary aim is to demonstrate the feasibility and advantages of the instruction-tuning paradigm for EEG tasks, and we look forward to exploring these enhancements to further unlock the potential of NeuroLM in practical applications.
>
> > **[W2 & Q1]** Dependency on GPT-2
> - Thank you for this feedback. We appreciate the interest in seeing NeuroLM evaluated with more advanced LLMs. In this work, we used GPT-2, a smaller and older LLM, to establish the feasibility of our approach. The fact that NeuroLM performs well with GPT-2 gives us confidence that it would achieve even stronger results with more powerful models, such as LLaMA series or other recent LLMs designed for enhanced performance and multimodal capabilities. Due to limited computational resources and the high demands of training larger LLMs, we apologize that we are unable to add comparisons with other models in this rebuttal stage. However, exploring NeuroLM’s performance with advanced LLMs is an exciting avenue for future work, and we believe that our method’s adaptability across models will offer even broader potential as it scales with more sophisticated LLM architectures. Anyway, we hope to present our preliminary results in this paper to inspire future exploration.
>
> > **[W3]** EEG-Text Alignment
> - To assess the impact of EEG-text alignment, we have conducted ablation studies in which we removed the alignment loss, and the results showed that training fails entirely without it. Specifically, without the alignment, the model struggled to produce meaningful predictions, outputting arbitrary codes instead of expected responses like (A), (B), or \(C), resulting in near-zero scores across most metrics. This appears to stem from disordered attention scores, as EEG and text embeddings remain unaligned in separate spaces. We have detailed these observations in Appendix F.2, and we appreciate your suggestion, as it highlights the importance of EEG-text alignment in NeuroLM's overall performance and robustness.
>
> > **[W4]** Complexity and Reproducibility
> - We promise to make our codes and model weights publicly available based on publication for the community to further explore this new paradigm.
>
> > **[Q2]** What role does adversarial training play...
> - In this work, we used adversarial training for EEG-text alignment to demonstrate the feasibility of our new multi-task instruction-tuning paradigm with LLMs. This approach provides a straightforward way to align EEG signals with text embeddings, enabling NeuroLM to process multiple EEG tasks within an LLM framework. While adversarial training was effective for this initial demonstration, we recognize that contrastive learning-based alignment could further enhance model performance by achieving more detailed alignment between EEG and text as mentioned in Appendix H. However, incorporating contrastive learning would introduce significant complexity, potentially detracting from our focus on establishing the feasibility of this multi-task framework. We see contrastive alignment as a promising direction for future work, allowing us to refine and advance the alignment mechanism in NeuroLM.

---

> > ### Author Response · Authors · 2024-11-25
> >
> > Thank you for your thoughtful and insightful suggestions. We believe we have comprehensively addressed your questions regarding NeuroLM’s performance with fine-tuning, its dependency on GPT-2, the EEG-text alignment, and its reproducibility.
> >
> > We would like to emphasize that our method is the first to effectively integrate multiple tasks into a single model by leveraging the question-answering paradigm of LLMs in a true multi-task manner. NeuroLM achieves competitive performance even with a relatively simple GPT-2 backbone and our effective EEG-text alignment strategy, demonstrating the robustness and potential of our approach. We believe this pioneering work lays a strong foundation for future advancements in EEG-Language integration.
> >
> > We are wondering whether you have any additional questions or comments regarding our response to your review comments. We will do our best to address them.
> >
> > We sincerely appreciate the time and effort you have dedicated to reviewing our manuscript. Thank you for your thoughtful consideration!

---

> > > ### Comment · Reviewer_hyNm · 2024-11-25
> > >
> > > Thank you for your detailed response to my review.
> > >
> > > - The inclusion of a discussion on future directions (e.g., leveraging stronger LLMs, improving EEG-text alignment with contrastive learning) demonstrates an awareness of the work’s limitations and opportunities for improvement.
> > >
> > > - Your justification for using GPT-2 due to resource constraints is valid, and I appreciate your acknowledgment of the need to evaluate NeuroLM on more advanced LLMs.
> > >
> > > - The ablation results substantiate the importance of adversarial training in the current model design, which is an acceptable first step toward demonstrating feasibility.
> > >
> > >
> > > While your responses effectively address most of the weaknesses, the lack of evaluation with more advanced LLMs remains a limitation. As you pointed out, future work will address this, but the current study could have benefited from even limited experiments on more modern LLMs to validate scalability and robustness.
> > > The complexity of the model pipeline, while partially mitigated by your promise of open-sourcing, may still pose challenges for adoption by the broader research community.
> > >
> > > Based on your answers, I am inclined to increase the score from 5 to 6. While the study has limitations, the novelty of the approach makes a valuable contribution to the field.

---

> > > > ### Author Response · Authors · 2024-11-25
> > > >
> > > > Thank you for your recognition of our paper and the improved score! We also appreciate the valuable suggestions you provided.

---

### Official Review · Reviewer_2GzD · 2024-11-03

**Soundness:** 3
**Presentation:** 3
**Contribution:** 3
**Rating:** 5
**Confidence:** 3

**Summary:**

This paper introduces a multi-task foundation model called NeuroLM, which is integrated into large language models (LLMS) for electroencephalogram (EEG) signal processing. NeuroLM uses a text-aligned neural tokenizer to encode continuous EEG signals into discrete neural labels by learning through vector-quantized temporal-frequency prediction. These labels are subsequently used as input to the LLM through multi-channel autoregressive pre-training, which enables the model to learn a causal representation of EEG signals. Finally, through multi-task instruction tuning, NeuroLM is able to handle multiple downstream tasks, such as anomaly detection, event type classification, and emotion recognition, in a single model.

**Strengths:**

1. NeuroLM demonstrates for the first time the ability of a model to handle multiple EEG tasks simultaneously with instruction tuning.
2. NeuroLM is evaluated on six different downstream datasets, demonstrating its broad applicability and superior learning capabilities.
3. NeuroLM reduces the need for repeated training through instruction adjustment, making the model more efficient.

**Weaknesses:**

1. While NeuroLM performs well in multi-task learning, there are still gaps in its performance on some tasks compared to state-of-the-art models specifically designed and optimized for a single task (e.g., LaBraM).
2. Model performance fluctuates greatly across different tasks and datasets.
3. Although the model reduces the need for a large amount of labeled data through pre-training and multi-task learning, its performance and generalization ability still rely on high-quality training data.

**Questions:**

1. The paper mentions that NeuroLM performs well on several EEG tasks, but is there any evaluation of how well the model generalizes across different datasets?
2. How sensitive is the model to noisy data and how does it perform under varying degrees of EEG signal quality variation?

---

> ### Author Response · Authors · 2024-11-19
> **Response (1/2)**
>
> Thanks for your wisdom and valuable comments. Here are our responses to your questions and concerns point-by-point.
> > **[W1]** While NeuroLM performs well in multi-task learning...
> - **Our target:** The primary objective of this work is to enable off-the-shelf LLMs to perform EEG tasks through a unified instruction-tuning paradigm. NeuroLM’s key strengths lies in its unified instruction-tuning approach, which may have the potential to enable generalization to novel tasks or prompts without the need for extensive task-specific fine-tuning. This work is the first to demonstrate that such a multi-task instruction-tuning approach can be effective for EEG tasks within an LLM framework.
> - **Reasons for not achieving SOTA:** NeuroLM's multi-channel autoregressive pre-training approach was selected to ensure compatibility with the causal structure of current LLMs, even though it may not reach the efficiency of masked signal modeling, which is known to produce strong results in single-task models like LaBraM. The autoregressive approach allows us to adapt EEG data to a causal language model framework, enabling multi-task learning within the constraints of current LLM architectures.
> - **Potential improvements:** We do recognize opportunities to further enhance NeuroLM’s performance. As mentioned in Appendix H, future improvements could involve contrastive learning to align EEG signals with textual descriptions in the VQ encoder, similar to the CLIP framework. This alignment would allow a more nuanced integration of EEG and text data and could yield significant gains in task performance.
> In this work, our focus was on establishing the feasibility of integrating EEG with LLMs in a unified, multi-task framework. Adding contrastive learning to our pre-training process would introduce substantial complexity, so we view this as a promising direction to pursue in future work. That said, we aim to present our preliminary results in the paper to encourage further exploration in this area.
>
> > **[W2]** Model performance fluctuates...
> - Thank you for noting this point. We observed that performance variability across tasks and datasets is influenced by several factors, including dataset size, task complexity, and variability in EEG data quality. For example, tasks with more extensive and balanced datasets allow NeuroLM to capture more representative features, resulting in stable performance. In contrast, smaller or highly specialized datasets tend to introduce variability, as the model’s generalization capacity is challenged. It’s worth noting that, apart from LaBraM, NeuroLM achieves comparable or even superior performance relative to other single-task baselines across most downstream datasets, underscoring the effectiveness of its unified multi-task learning approach. To further stabilize performance across datasets, one promising direction for future work involves exploring adaptive weighting strategies. This approach could help balance learning across tasks with different data distributions and complexities.
>
> > **[W3]** Although the model reduces the need...
> - Thank you for raising this point. While NeuroLM’s pre-training and multi-task learning approaches help reduce dependence on extensive labeled data, the model’s performance and generalization are indeed influenced by the quality of its training data. High-quality EEG data is crucial for capturing robust and generalizable representations, as it helps mitigate noise and variability that can affect model effectiveness—a common challenge across EEG processing methods. Many datasets contain some lower-quality samples, such as those with interference due to subject movement (EMG, EOG). Such noise can detract from model performance. For a fair comparison, we included all data across methods without filtering, but removing low-quality data presents a promising opportunity to further enhance NeuroLM’s performance.
>
> > **[Q1]** The paper mentions that NeuroLM...
> - Thank you for this question. NeuroLM’s unified instruction-tuning approach enables it to handle multiple EEG tasks across diverse datasets without requiring task-specific fine-tuning. Notably, we did not include the specific downstream datasets during neural tokenizer training and multi-task instruction tuning. In our experiments, we observe that NeuroLM maintains competitive performance across six distinct datasets, suggesting robust generalization. However, we acknowledge that a systematic evaluation of cross-dataset generalization could provide additional insights. In future work, we plan to evaluate NeuroLM’s transferability explicitly by testing on new datasets, further assessing its adaptability and robustness across varied EEG datasets.

---

> > ### Author Response · Authors · 2024-11-19
> > **Response (2/2)**
> >
> > > **[Q2]** How sensitive is the model...
> > - Thank you for this question. Current EEG denoising methods often rely on synthetic noisy data, such as adding EMG or EOG signals to clean EEG samples, to evaluate models’ robustness to noise [1]. In contrast, our focus in this work was not on handling noisy EEG data specifically, but rather on developing a unified multi-task instruction-tuning framework with NeuroLM. Our primary aim was to demonstrate NeuroLM’s adaptability and versatility across a variety of EEG tasks without requiring task-specific fine-tuning.
> >
> > [1] Zhang H, Zhao M, Wei C, et al. EEGdenoiseNet: a benchmark dataset for deep learning solutions of EEG denoising. Journal of Neural Engineering, 2021.

---

> > > ### Author Response · Authors · 2024-11-25
> > >
> > > Thank you for your thoughtful and insightful suggestions. We believe we have comprehensively addressed your questions regarding NeuroLM’s performance, its handling of noisy data, and its generalization ability.
> > >
> > > We would like to emphasize that our method is the first to effectively integrate multiple tasks into a single model by leveraging the question-answering paradigm of LLMs in a true multi-task manner. NeuroLM demonstrates competitive performance even with significantly varied data sizes across different tasks, showcasing the robustness and promise of our approach. We believe this pioneering work lays a strong foundation for future advancements in EEG-Language integration.
> > >
> > > We are wondering whether you have any additional questions or comments regarding our response to your review comments. We will do our best to address them.
> > >
> > > We sincerely appreciate the time and effort you have dedicated to reviewing our manuscript. Thank you for your thoughtful consideration!

---

### Official Review · Reviewer_4vcv · 2024-11-04

**Soundness:** 4
**Presentation:** 4
**Contribution:** 3
**Rating:** 8
**Confidence:** 4

**Summary:**

The paper presents NeuroLM, an innovative multi-task foundation model that bridges the gap between electroencephalogram (EEG) signals and language models. By converting EEG signals into discrete tokens aligned with text embeddings, NeuroLM enables Large Language Models (LLMs) to process EEG data effectively. The model employs a neural tokenizer trained through vector-quantized temporal-frequency prediction to encode EEG signals, and integrates these tokens into an LLM using multi-channel autoregressive pre-training. Through multi-task instruction tuning, NeuroLM adapts to various downstream EEG tasks without requiring task-specific fine-tuning. Evaluated on six diverse EEG datasets, NeuroLM demonstrates its potential to unify multiple EEG tasks within a single model.

**Strengths:**

1. The paper presents an innovative method that enables handling multiple diverse EEG tasks within a single unified model without the need for task-specific fine-tuning. the authors advance the capability to perform multi-task learning in EEG signal processing, overcoming limitations of traditional models that require separate fine-tuning for each task.

2. The authors conduct thorough experiments on six diverse downstream EEG tasks, including abnormal detection, event type classification, emotion recognition, sleep stage classification, cognitive workload detection, and slowing event classification. This extensive evaluation provides a robust comparison with existing state-of-the-art models, demonstrating the versatility and potential of NeuroLM across different EEG applications.

3. The paper includes detailed ablation studies examining the effectiveness of various components of the proposed model, such as the EEG-text embedding alignment, multi-channel autoregressive pre-training, and the robustness to instruction variations. These analyses validate the contributions of each component, offer insights into the model's performance, and enhance the understanding of how each part contributes to the overall effectiveness.

4. The authors provide clear and detailed descriptions of the model architecture, training procedures, data preprocessing steps, and experimental setups. This transparency facilitates reproducibility and allows other researchers to replicate and build upon the work more easily.

**Weaknesses:**

Overall the paper has a good contribution and is well structured and clear. Here are some tips for improvements:

1. Even though NeuroLM's performance is currently lower compared to LaBraM on most tasks, the authors could emphasize the benefits brought by the unified instruction tuning approach. Specifically, since NeuroLM is instruction-tuned, it may exhibit better generalization to unseen tasks or prompts, enabling zero-shot or few-shot learning without additional fine-tuning. This generalizability could make NeuroLM more adaptable to new EEG tasks compared to models like LaBraM that require task-specific fine-tuning, thus presenting a significant advantage of their approach.

2. The section on methodology especially the codebook and its origin from VQ-VAE,  and how it functions within NeuroLM, and the stop gradient operator justification in equation (3), could benefit from some explanation.

3. For the EEG-text alignment, while an appendix mentions visualization, a more detailed analysis quantifying the impact of EEG-text alignment on model performance would strengthen the claims.

4. Change the colormap for Figure 1, currently the color for NeuroLM and LaBraM are very similar and hard to distinguish.

**Questions:**

1. It would be highly beneficial for the research community if you could share the code for NeuroLM as supplementary material.

2. The expected benefits of large-scale pretraining are not fully realized on the TUAB dataset, with only minimal improvement observed. I was anticipating that the extensive pretraining would lead to more significant performance gains. Could you provide insights into why the pretraining did not result in better performance on TUAB?

---

> ### Author Response · Authors · 2024-11-19
> **Response**
>
> Thanks for your wisdom and valuable comments. We sincerely appreciate your recognition of the contribution of our work. Your tips are really helpful for improving the quality of our paper. Here are our responses to your questions and concerns point-by-point.
> > **[W1]** Even though NeuroLM's performance is currently lower compared to LaBraM...
> - We are deeply grateful for your suggestion that NeuroLM's key strengths lies in its unified instruction-tuning approach, which indeed has the potential to enable generalization to novel tasks or prompts without the need for extensive task-specific fine-tuning. We have expanded on this in the revised manuscript in Section 3.3 by highlighting that NeuroLM’s instruction-tuned design can facilitate zero-shot or few-shot generalization to unseen EEG tasks, as it has been optimized to handle a broad range of instructions and queries.
>
> > **[W2]** The section on methodology...
> - Thank you for highlighting the need for additional clarification regarding the VQ-VAE codebook and its role in NeuroLM. The VQ-VAE framework is essential for discretizing continuous EEG signals into neural tokens, which we then align with LLM text embeddings. To initialize the codebook, we use k-means clustering on the initial batch of embeddings from the VQ encoder, creating 8192 cluster centers that form the basis of the 8192 codes in the codebook.
> - For codebook learning, we follow Equation (3) to iteratively update the codes. Since the lookup and replacement operations are undifferentiable, we apply a stop-gradient operation to transfer gradients from the codebook directly to the VQ encoder, effectively stabilizing the training. The second term in Equation (3) aligns the codes with embeddings from the VQ encoder, while the third term brings these embeddings closer to the codes in the codebook. After training the neural tokenizer, the codebook is merged with the LLM’s text vocabulary, enabling multi-channel autoregression.
> - Since this tokenizer training strategy aligns closely with LaBraM, we simplified its explanation in the initial draft. We apologize for any inconvenience and will include these detailed explanations in our revision.
>
> > **[W3]** For the EEG-text alignment...
> - Thanks for your suggestion. We have conducted experiments removing the alignment loss, and the results confirm that training fails entirely without it. Without alignment, the model struggles to produce meaningful predictions, often outputting arbitrary codes instead of expected options like (A), (B), or \(C), leading to near-zero scores across most metrics on downstream tasks. This outcome appears to stem from disordered attention scores, as EEG and text embeddings remain in separate spaces. We have detailed this observation in Appendix F.2.
>
> > **[W4]** Change the colormap for Figure 1...
> - We have revised the colormap accordingly to ensure a clear distinction between the models, enhancing readability.
>
> > **[Q1]** It would be highly beneficial...
> - Definitely, we will make our codes and model weights publicly available based on publication.
>
> > **[Q2]** The expected benefits of large-scale pretraining...
> - Thank you for highlighting this important point. Frankly, the choice of multi-channel autoregressive pre-training in NeuroLM was indeed a compromise. It is widely acknowledged that masked signal modeling tends to be a more efficient representation learning approach than next-token prediction, which contributes to LaBraM’s stronger performance. Given the causal nature of current LLMs, we aimed to find a feasible pre-training strategy that works within these constraints, with autoregressive modeling being the most compatible option for EEG tokens.
> However, NeuroLM does have potential pathways for performance improvements. As noted in Appendix H, one promising direction involves leveraging contrastive learning to align EEG signals with textual descriptions within the VQ encoder, drawing inspiration from approaches like CLIP [1]. This would enable more detailed alignment with text and likely bring significant performance gains to NeuroLM.
> Nonetheless, our current work focuses on establishing the feasibility of incorporating EEG into LLMs through the multi-task instruction-tuning framework. Introducing contrastive learning for pre-training, while promising, would add considerable complexity to the current study, and we see this as an exciting avenue for future work.
>
> [1] Radford A, Kim J W, Hallacy C, et al. Learning transferable visual models from natural language supervision. ICML 2021

---

### Author Response · Authors · 2024-11-19
**Global response**

We would like to express our sincere gratitude to all reviewers for their insightful and constructive feedback on our work.

### Focus and Key Contribution
Our goal is to **lay the foundation for a new multi-task paradigm in EEG processing using LLMs**, demonstrating that a unified instruction-tuning approach can effectively handle diverse EEG tasks. We're glad to find that all reviewers recognized NeuroLM’s unique contribution in this regard, particularly its ability to generalize across multiple EEG tasks within a single model. This approach enables NeuroLM to generalize across a range of tasks and we are delighted to see it highlighted by reviewer 4vcv as a promising direction for expanding the application of LLMs in EEG-based research.

Moreover, we appreciate that reviewer hyNm and gQYU pointed out some possible directions for improvement and reviewer 2GzD and hyNm provided comments on concrete targets to achieve this whole branch of work. We envision NeuroLM as a stepping stone for more advanced EEG-Language integration models, and we hope that this work will inspire future research to refine and expand on this framework. We are excited about the potential for more sophisticated alignment techniques, multi-task learning frameworks, and robust training strategies to build on NeuroLM’s foundation, contributing to broader applications of LLMs in EEG and neuroscience.

### Common Concerns and Clarifications
- **Performance of NeuroLM** Several reviewers noted areas for potential improvement and clarification. We acknowledge that NeuroLM does not achieve state-of-the-art (SOTA) performance on every task, particularly when compared to specialized, single-task models like LaBraM. We also want to emphasize that, unlike many so-called multi-task models that need specific components for each task, **NeuroLM is a true multi-task model**, meaning that the same trained model executes multiple tasks simultaneously without any additional module. This inherently introduces a trade-off problem between the performance of different tasks, as the model must generalize across diverse datasets and objectives without task-specific fine-tuning. Despite this trade-off problem in unified models, NeuroLM achieves competitive performance on multiple tasks, demonstrating the effectiveness of this paradigm. Besides, the choice of multi-channel autoregressive pre-training was a necessary compromise to ensure compatibility with the causal structure of current LLM for this study. It is well known that masked signal modeling can be a more efficient representation learning approach than next-token prediction, which contributes to LaBraM’s stronger performance. In contrast, next-token prediction appears to be better for generation and has a better scaling property. Given the causal nature of current LLMs, we aimed to find a feasible pre-training strategy that works within these constraints, with autoregressive modeling being the most compatible option for EEG tokens.
We believe that incorporating contrastive learning is a promising direction for improving EEG-text alignment. As noted in Appendix H, contrastive learning could help align EEG signals with text descriptions within the VQ encoder, inspired by frameworks like CLIP. This alignment would allow for more nuanced EEG-text integration and potentially yield significant performance gains. Introducing contrastive learning for pre-training, while promising, would add considerable complexity to the current study, and we see this as an exciting avenue for future work.
- **Ablation on EEG-text alignment** Reviewers also requested further clarification on the impact of EEG-text alignment. To assess this, we conducted ablation studies by removing the alignment loss, and the results showed that training fails entirely without it. Specifically, without alignment, the model struggled to produce meaningful predictions and outputted random codes instead of expected responses like (A), (B), or \(C), leading to near-zero scores across most metrics. This misalignment likely results from disordered attention scores, as EEG and text embeddings remain unaligned in separate spaces. We have detailed these findings in Appendix F.2, which you may have missed.

---

> ### Author Response · Authors · 2024-11-19
> **Global response (part 2)**
>
> ### Paper Revisions and Additions
> In response to reviewer feedback, we have revised several areas of the paper to enhance clarity and address specific points:
>
> 1.	We clarified visualizations of EEG-text embedding alignment in Appendix F.2, illustrating the embeddings before and after alignment.
> 2.	We highlight NeuroLM’s key contribution when comparing to other single-task baselines in Section 3.3.
> 3.	We added details regarding the specific text embeddings used during EEG-text alignment in Section 3.2.
> 4.	We adjusted the color for LaBraM in Figure 1 to improve visualization clarity.
> 5.	We modified the Theory Analysis section in 2.2 to provide a more detailed formulation.
>
> ### Long-Term Contribution
> With this paper, we hope to make a lasting contribution to the academic community by demonstrating the feasibility of using LLMs for multi-task EEG instruction tuning. We see NeuroLM as the first step in incorporating EEG into LLMs for multi-task question-answering paradigms, and we hope this work will serve as a foundation for future research to further enhance EEG-LLM integration.
>
> Once again, thank you for your valuable feedback, which has greatly strengthened our work.

---

### Author Response · Authors · 2024-11-23
**Inquiry for post-rebuttal comments**

Dear Reviewers:

Thank you again for your wisdom and valuable comments. We have provided complete explanations for all the questions. Since the rebuttal process is approaching its end, we would be glad to hear from you whether our rebuttal has addressed your concerns. Feel free to comment on our rebuttal if you have further questions or considerations.

---

### Meta-Review · Area_Chair_qNBJ · 2024-12-16

**Metareview:**

The paper presents an innovative method for predicting on diverse EEG tasks within a single unified model without the need for task-specific fine-tuning.

Pros:
- The approach is considered original and novel
- Models are trained on a very large EEG corpus
- Results are provided on diverse set of tasks
- Paper provides an ablation study

Minor concerns:
- Code is not shared and given the complexity of the pipeline reproducibility of the results would certainly be a big challenge for the field. Yet authors promise to release code and weights upon acceptance. Please make sure that the code allows full replication of the results.
- Remaining experiments (impact of LLM, instructions etc.) should be considered but realistically in future works.

As the AC, I would strongly encourage this community to collaborate in order to facilitate benchmarks and comparisons of methods on standard tasks with minimal code changes.

**Additional Comments On Reviewer Discussion:**

Reviewer remains non-supportive of the work but the review is rather superficial and 2GzD did not engage with the authors during the discussion period. Other reviewers acknowledge the merit and novelty of the method endorsing the work for publication.

---

### Decision · Program_Chairs · 2025-01-22

Accept (Poster)